# Influences of hydroxyl radicals (OH) on top-down estimates of the global and regional methane budgets

Yuanhong Zhao[1], Marielle Saunois[1], Philippe Bousquet[1], Xin Lin[1,a], Antoine Berchet[1], Michaela I. Hegglin[2], Josep G. Canadell[3], Robert B. Jackson[4], Edward J. Dlugokencky[5], Ray L. Langenfelds[6], Michel Ramonet[1], Doug Worthy[7], and Bo Zheng[1]

[1] Laboratoire des Sciences du Climat et de l'Environnement, LSCE-IPSL (CEA-CNRS-UVSQ), Université Paris-Saclay, 91191 Gif-sur-Yvette, France
[2] Department of Meteorology, University of Reading, Reading, RG6 6LA, UK
[3] Global Carbon Project, CSIRO Oceans and Atmosphere, Canberra, Australian Capital Territory 2601, Australia
[4] Earth System Science Department, Woods Institute for the Environment, and Precourt Institute for Energy, Stanford University, Stanford, CA 94305, USA
[5] NOAA ESRL, 325 Broadway, Boulder, CO 80305, USA
[6] Climate Science Centre, CSIRO Oceans and Atmosphere, Aspendale, Victoria 3195, Australia
[7] Environment and Climate Change Canada, Toronto, M3H 5T4, Canada

[a] now at: Climate and Space Sciences and Engineering, University of Michigan, Ann Arbor, MI 48109, USA

**Abstract**

The hydroxyl radical (OH), which is the dominant sink of methane ($CH_4$), plays a key role to close the global methane budget. Current top-down estimates of the global and regional $CH_4$ budget using 3D models usually apply prescribed OH fields and attribute model-observation mismatches almost exclusively to $CH_4$ emissions, leaving the uncertainties due to prescribed OH field less quantified. Here, using a variational Bayesian inversion framework and the 3D chemical transport model LMDz, combined with 10 different OH fields derived from chemistry-climate models (CCMI experiment), we evaluate the influence of OH burden, spatial distribution, and temporal variations on the global and regional $CH_4$ budget. The global tropospheric mean $CH_4$-reaction-weighted [OH] ($[OH]_{GM-CH4}$) ranges $10.3\text{-}16.3 \times 10^5$ molec $cm^{-3}$ across 10 OH fields during the early 2000s, resulting in inversion-based global $CH_4$ emissions between 518 and 757 Tg $yr^{-1}$. The uncertainties in $CH_4$ inversions induced by the different OH fields are similar to the $CH_4$ emission range estimated by previous bottom-up syntheses and larger than the range reported by the top-down studies. The uncertainties in emissions induced by OH are largest over South America, corresponding to large inter-model differences of [OH] in this region. From the early to the late 2000s, the optimized $CH_4$ emissions increased by $21.9 \pm 5.7$ Tg $yr^{-1}$ (16.6-30.0 Tg $yr^{-1}$), of which ~25% (on average) offsets the 0.7% (on average) increase in OH burden. If the CCMI models represent the OH trend properly over the 2000s, our results show that a higher increasing trend of $CH_4$ emissions is needed to match the $CH_4$ observations compared to the $CH_4$ emission trend derived using constant OH. This study strengthens the importance to reach a better representation of OH burden and of OH spatial and temporal distributions to reduce the uncertainties on the global and regional $CH_4$ budgets.

## 1 Introduction

Methane ($CH_4$) plays an important role in both climate change and air quality as a major greenhouse gas and tropospheric ozone precursor (Ciais et al., 2013). $CH_4$ is emitted from various anthropogenic sources including agriculture, waste, fossil fuel, and biomass burning, as well as natural sources including wetlands and other freshwater systems, geological sources, termites, and wild animals. $CH_4$ is removed from the atmosphere mainly by reaction with hydroxyl radical (OH) (Saunois et al. 2016, 2017). Tropospheric $CH_4$ levels have more than doubled between the 1850s and present-day (Etheridge et al., 1998) in response to anthropogenic emissions and climate variabilities, leading to about 0.62 W m$^{-2}$ radiative forcing (Etminan et al., 2016) and increases in tropospheric ozone levels of ~5 ppbv (Fiore et al., 2008). The global $CH_4$ atmospheric mixing ratio stabilized in the early 2000s but resumed growing at a rate of ~5ppbv yr$^{-1}$ or more since 2007 (Dlugokencky, NOAA/ESRL, 2019).

Explaining the $CH_4$ stabilization and renewed growth requires an accurate estimation of the $CH_4$ budget and its evolution, as the source-sink imbalance that is responsible for the contemporary $CH_4$ yearly growth only accounts for 3% of the total $CH_4$ burden (Turner et al., 2019). To reconcile the uncertainties in the current estimation of $CH_4$ emissions from various sources, the Global Carbon Project integrates top-down and bottom-up approaches (Kirschke et al. 2013; Saunois et al., 2016; 2017; 2019). However, gaps remain in global and regional $CH_4$ emissions estimated by top-down and bottom-up approaches, as well as within each approach (Kirschke et al. 2013; Saunois et al., 2016; Bloom et al., 2017). The top-down method, which optimizes emissions by assimilating observations in an atmospheric inversion system, is expected to reduce uncertainties of bottom-up estimates. Among the remaining causes of uncertainties in the global methane budget, the representation of $CH_4$ sinks, mainly from OH oxidation, is one of the largest, as seen by process-based models for atmospheric chemistry (Saunois et al., 2017).

OH is the most important tropospheric oxidizing agent determining the lifetime of many pollutants and

greenhouse gases including $CH_4$ (Levy, 1971). A small perturbation of OH can result in significant changes in the budget of atmospheric $CH_4$ (Turner et al., 2019). At the global scale, tropospheric OH is mainly produced by the reaction of excited oxygen atoms ($O(^1D)$) with water vapor (primary production) but also by the reaction of nitrogen oxide (NO) and ozone ($O_3$) with hydroperoxyl radicals ($HO_2$) and organic peroxy radicals ($RO_2$) (secondary production). At regional scales, photolysis of hydrogen peroxide and oxidized VOC photolysis can be important depending on the chemical environment (Lelieveld et al. 2016). OH is rapidly removed by carbon monoxide (CO), methane ($CH_4$), and non-methane volatile organic compounds (NMVOCs) (Logan et al., 1981; Lelieveld et al., 2004). Tropospheric OH has a very short lifetime of a few seconds (Logan et al., 1981; Lelieveld et al., 2004), hindering estimates of global OH concentrations ([OH]) through direct measurements and limiting our ability to estimate the global $CH_4$ sink.

Global tropospheric [OH] is approximately $1 \times 10^6$ molec $cm^{-3}$ as calculated by atmospheric chemistry models (Naik et al., 2013; Voulgarakis et al., 2013, Zhao et al., 2019) and inversions of 1-1-1trichloroethane (methyl chloroform, MCF) (Prinn et al., 2001; Bousquet et al., 2005; Montzka et al., 2011; Cressot et al., 2014), resulting in a chemical lifetime of ~9 years for tropospheric $CH_4$ (Prather et al., 2012; Naik et al., 2013). However, accurate estimation of [OH] magnitude, distributions, and year-to-year variations needed for $CH_4$ emission optimizations are still pending (Prather et al., 2017; Turner et al., 2019). For global tropospheric [OH], both MCF inversions and atmospheric chemistry model inter-comparisons give a 10%-15% uncertainty (Prinn et al., 2001; Bousquet et al., 2005; Naik et al., 2013; Zhao et al., 2019). For [OH] spatial distributions, MCF-based inversions generally infer similar mean [OH] over both hemispheres (Bousquet et al., 2005; Patra et al., 2014), while atmospheric chemistry models generally give [OH] Northern hemisphere to Southern hemisphere (N/S) ratios above 1 (e.g. Naik et al., 2013; Zhao et al., 2019). For [OH] year-to-year variations, some studies have estimated magnitudes significant enough to help explain part of the stagnation in atmospheric $CH_4$ concentrations during the

early 2000s (Rigby et al., 2008; McNorton et al., 2016; Dalsøren et al., 2016; Rigby et al., 2017; Turner et al., 2017), whereas others show smaller trends and inter-annual variations of [OH] (Montzka et al, 2011; Naik et al., 2013; Voulgarakis et al., 2013; Zhao et al., 2019). In a recent study, Zhao et al. (2019) simulated atmospheric $CH_4$ with an ensemble of OH fields and showed that uncertainties in [OH] variations could explain up to 54% of model-observation discrepancies in surface $CH_4$ mixing ratio changes from 2000 to 2016.

Current top-down estimates of the global $CH_4$ budget usually apply prescribed and constant [OH] simulated by atmospheric chemistry models and attribute model-observation mismatches exclusively to $CH_4$ emissions (Saunois et al., 2017). However, the OH fields simulated by atmospheric chemistry models show some uncertainties in both global burden and spatial-temporal variations (Naik et al., 2013; Zhao et al., 2019). The role of OH variations on the top-down estimates of $CH_4$ emissions has been evaluated using two box-model inversions with surface observations (e.g. Rigby et al., 2017; Turner et al., 2017, Naus et al., 2019) and 3D models that optimize $CH_4$ emissions together with [OH] by assimilating surface observations (Bousquet et al., 2006) or satellite data (Cressot et al., 2014, McNorton et al., 2018; Zhang et al., 2018; Maasakkers et al., 2019). The proxy-based constraints usually optimize [OH] on a global or latitudinal scale, the impact of OH vertical and horizontal distributions being less quantified to date. Also, proxy methods do not allow to access underlying processes as direct chemistry modeling (Zhao et al., 2019). This paper follows the work of Zhao et al. (2019), where we analyzed in details 10 OH fields derived from atmospheric chemistry models considering different chemistry, emissions, and dynamics (Patra et al., 2011; Szopa et al., 2013; Hegglin and Lamarque, 2015; Morgenstern et al., 2017; Zhao et al., 2019; Terrenoire et al., 2019). We now aim to build on this previous paper to estimate the impact of these OH fields on methane emissions as inferred by an atmospheric 4D variational inversion system. To do so, we use each of the OH fields in the 4D variational inversion system PYVAR-LMDz based on LMDZ-SACS (Laboratoire de Météorologie Dynamique model with Zooming capability-Simplified Atmospheric

Chemistry System) 3D chemical transport model to evaluate the influence of OH distributions and variations on the top-down estimated global and regional $CH_4$ budget. Section 2 briefly describes the OH fields and their characteristics and underlying processes (see also Zhao et al., 2019 for more details), the inversion method, and the setups of inversion experiments. Section 3 illustrates the influence of OH on the top-down estimation of $CH_4$ budgets and variations, specifically: i) global, regional, and sectoral $CH_4$ emissions (Section 3.1), ii) emission changes between the early 2000s and late 2000s (Section 3.2), and iii) year-to-year variations in methane emissions (Section 3.3). Section 4 summarizes the results and discusses the impact of OH on the current $CH_4$ budget.

## 2 Method

### 2.1 OH fields

In this study, we test the 10 OH fields presented in by Zhao et al. (2019), including 7 OH fields simulated by chemistry-transport and chemistry-climate models from Phase 1 of the Chemistry-Climate Model Initiative (CCMI) (Hegglin and Lamarque, 2015; Morgenstern et al., 2017), 2 OH fields simulated by the Interaction with Chemistry and Aerosols (INCA) model coupled to the general circulation model of the Laboratoire de Météorologie Dynamique (LMD) model (Hauglustaine et al., 2004; Szopa et al., 2013), and 1 OH field from the TRANSCOM-CH4 inter-comparison exercise (Patra et al., 2011)(Table 1).

The CCMI project conducted simulations with 20 state-of-the-art atmospheric chemistry-climate and chemistry-transport models to evaluate the model's projections of atmospheric composition (Hegglin and Lamarque, 2015; Morgenstern et al., 2017). To force atmospheric inversions during 2000-2010, we use OH fields from 7 of the 20 CCMI model simulations of REF-C1 experiments (Table 1), which were driven by observed sea surface temperatures and state-of-the-art historical forcings (covering 1960-2010). For the inversions after 2010 (only with the CESM1-WACCM model, see Section 2.3), we apply inter-annual variations of OH generated from REF-C2 experiments, which were driven by sea surface conditions

calculated by the online-coupled ocean and sea ice modules. Although all of the CCMI models use the same anthropogenic emission inventories, the simulated OH fields show different spatial and vertical distributions. The inter-model differences of OH burden and vertical distributions are mainly attributed to differences in chemical mechanisms related to NO production and loss. The differences in [OH] spatial distributions are due to applying different natural emissions: for example, primary biogenic VOC emissions and NO emissions from soil and lightning (Zhao et al., 2019). As a result, the regions dominated by natural emissions (e.g. South America, central Africa) show the largest inter-model differences in [OH] (Fig.S1). The CCMI models consistently simulated positive OH trend during 2000-2010, mainly due to more OH production by NO than loss by CO over the East and Southeast Asia and positive trend of water vapor over the tropical regions (Zhao et al., 2019; Nicely et al., 2020). More details can be found in Zhao et al. (2019) and the herein cited literature.

The two INCA OH fields, INCA NMHC-AER-S and INCA NMHC are simulated by two different versions of the INCA (Interaction with Chemistry and Aerosols) chemical model coupled to LMDz (Szopa et al., 2013; Terrenoire et al., 2019). The main difference between the two simulations is that INCA NMHC-AER-S includes both gas-phase and aerosol chemistry in the whole atmosphere while INCA NMHC only includes gas-phase chemistry in the troposphere (Szopa et al., 2013; Terrenoire et al., 2019). We also include the climatological OH field used in the TransCom simulations (Patra et al., 2011), which uses the semi-empirical, observation-based OH field computed by Spivakovsky et al. (2000) in the troposphere.

Table 1 summarizes the global tropospheric mean $CH_4$-reaction-weighted [OH] ([OH]$_{GM-CH4}$, [OH] weighted by reaction rate of OH with $CH_4$ ($K_{OH+CH4}$)$\times$dry air mass, Lawrence et al., 2001) and dry air mass-weighted [OH] ([OH]$_{GM-M}$), as well as inter-hemispheric ratios (N/S ratios) calculated with [OH]$_{GM-CH4}$ for the 10 OH fields used in this study. The tropopause height is assumed at 200hPa following Naik

et al. (2013) and the 3D temperature field used to compute [OH]$_{GM-CH4}$ is from ERA Interim re-analysis meteorology data (Dee et al, 2011). The volume-weighted [OH] was given by Zhao et al. (2019). The [OH]$_{GM-CH4}$ is a better indicator of the global atmospheric oxidizing efficiency for CH$_4$ than [OH]$_{GM-M}$ since the latter is insensitive to the CH$_4$+OH reaction rate increased with temperature (Lawrence et al., 2001). Both the mean value (12.3±3.8×10$^5$ molec cm$^{-3}$) and absolute range (10.3-16.3×10$^5$ molec cm$^{-3}$) of [OH]$_{GM-CH4}$ calculated for the 10 OH fields are larger than those of [OH]$_{GM-M}$ (11.4±2.8×10$^5$ molec cm$^{-3}$ and 9.4-14.4×10$^5$ molec cm$^{-3}$, respectively). This is mainly because MOCAGE and SOCOL3 OH fields show much higher [OH] near the surface than in the upper troposphere (Zhao et al., 2019). The inter-hemispheric OH ratios range from 1.0 to 1.5, larger than ones derived from MCF inversions (e.g. Bousquet et al., 2005; Patra et al., 2014), partly explained by the underestimation of CO in the northern hemisphere by atmospheric chemistry models (Naik et al., 2013). A comprehensive analysis of spatial and vertical distributions of these OH fields was presented in Zhao et al. (2019).

## 2.2 Inverse method

We conduct an ensemble of variational inversions of CH$_4$ budget that rely on Bayes' theorem (Chevallier et al., 2005) with the same set of atmospheric observations of CH$_4$ mixing ratios but different prescribed monthly mean OH fields as described in Sect. 2.1. A variational data assimilation system optimizes CH$_4$ emissions and sinks by minimizing the cost function J, defined as:

$$J(x) = \frac{1}{2}(x - x^b)^T \mathbf{B}^{-1}(x - x^b) + \frac{1}{2}(H(x) - y)^T \mathbf{R}^{-1}(H(x) - y) \qquad (1)$$

where $x$ is the control vector that includes total CH$_4$ emissions per 10 days at the model resolution of 3.75 °(in longitude)×1.85 °(in latitude) and initial conditions at longitudinal and latitudinal bands of 20 °×15 °; $x^b$ is the prior of the control vector $x$; y is the observation vector of observed CH$_4$ mixing ratios, here at the surface; and $H(x)$ represents the sensitivity of simulated CH$_4$ to emissions, for comparison with $y$. $\mathbf{B}$ and $\mathbf{R}$ represent the prior and observation error covariance matrix, respectively. The cost

function $J$ is minimized iteratively by the M1QN3 algorithm (Gilbert and Lemaréchal, 1989). We do not include sinks in the control vector $x$ but prescribe the different OH fields mentioned above.


Prior knowledge ($x^b$) on $CH_4$ emissions is provided by the Global Carbon Project (GCP, Saunois et al., 2019). The GCP emission inventory includes time-varying anthropogenic and fire emissions and climatology of the natural emissions. Global total $CH_4$ emissions of the GCP inventory are $511Tg\ yr^{-1}$ in 2000, increased to $562Tg\ yr^{-1}$ in 2010, and $581Tg\ yr^{-1}$ in 2016 (with soil uptake excluded). The soil uptake

of $CH_4$ is estimated to be $38Tg\ yr^{-1}$ with seasonal variations. Averaged over 2000-2016, the anthropogenic sources (including biofuel emissions, agriculture, and waste) and wetlands contribute 56% and 32% of total $CH_4$ emissions, respectively (Fig. S2). The prior information of emissions by sector in each grid cell is used to separate the total optimized $CH_4$ emissions into four broad categories: wetlands, agriculture and waste, fossil fuel, and other natural sources (biomass burning, termite, geological, and ocean emissions).

The spatial distributions of the prior emissions from the four categories averaged over 2000-2016 are shown in Fig. S2. A detailed description of the GCP emission inventory can be found in Zhao et al. (2019) and Saunois et al. (2019). The prior error of $CH_4$ fluxes is set to 100% of $x^b$, and the error correlation is calculated with a correlation length of 500km over land and 1000km over the oceans for $CH_4$ fluxes.

The vector of observations ($y$) is generated from surface measurements of the World Data Centre for Greenhouse Gases (WDCGG, https://gaw.kishou.go.jp/ ) through the WMO Global Atmospheric Watch (WMO-GAW) program. The surface measurements include both continuous time series of hourly afternoon observations and flask data. In total, 97 sites are included here, covering different time periods, including 68 sites from the Earth System Research Laboratory from the U.S. National Oceanic and

Atmospheric Administration (NOAA/ESRL, Dlugokencky et al. (1994)), 14 sites from the Laboratoire des Sciences du Climat et de l'Environnement (LSCE), 8 sites from Environment and Climate Change Canada (ECCC), 4 sites from the Commonwealth Scientific and Industrial Research Organisation (CSIRO,

Francey et al. (1999)), and 3 from the Japan Meteorological Agency (JMA: http://www.jma.go.jp/jma/indexe.html). The location of the sites is shown in Fig. S3.


Atmospheric $CH_4$ sensitivities to fluxes ($H(x)$) are simulated by LMDz5B, an offline version of the LMDz atmospheric model (Locatelli et al., 2015), which has been widely used for $CH_4$ studies (e.g. Bousquet et al., 2005; Pison et al., 2009; Lin et al., 2018; Zhao et al., 2019). LMDz5B is associated with the simplified chemistry module SACS (Pison et al., 2009), which calculates the $CH_4$ chemical sink using prescribed

4D OH and $O(^1D)$ fields. The $CH_4$ sink by reaction with chlorine is not considered in our LMDz model simulations. The deep convection is parametrized based on the Tiedtke (1989) scheme. Air mass fluxes simulated by the general circulation model LMDz with temperature and wind nudged to ERA Interim re-analysis meteorology data (Dee et al, 2011) are used to force the transport of chemical tracers in LMDz5B every 3 hours.


### 2.3 Model experiments

As shown in Fig. 1, we performed six groups of inversions (Inv1 to Inv6, 34 inversions in total). The impacts of OH on the top-down estimation of $CH_4$ emissions are comprehensively analyzed by comparing the inversion results within one group or between two different groups. We analyze the overall impacts

of differences in OH burden, spatial distribution, and temporal change on $CH_4$ emissions (colored boxes on the right in Fig. 1), and separate the impacts of OH spatial distribution and temporal variations (colored boxes on the left in Fig. 1). The results are presented and discussed in three sections as shown in different colors in Fig. 1.

We perform four groups of 3-year $CH_4$ inversion experiments using 6 to 10 OH fields (Inv1 to Inv4, Fig. 1), and two groups of 17-year $CH_4$ inversions from 2000 to 2016 (Inv5 and Inv6, Fig. 1) using only CESM1-WACCM OH fields. For the short-term inversions, the first and last six months are treated as

spin-up and spin-down periods and discarded from the following analyses (to avoid edge effect). Thus, we only analyze the results over 2000/07-2002/06 (i.e. the early 2000s) for Inv1 and Inv2 and 2007/07-2009/06 (i.e. the late 2000s) for Inv3 and Inv4. The early 2000s and the late 2000s represent the time periods with stagnant and resumed growth of atmospheric $CH_4$ mixing ratios, respectively. For the long-term inversions, we take a one-year spin-up and spin-down and analyze the 15-year results from 2001 to 2015.

The aim of Inv1, conducted for 2000-2002 with 10 OH fields, is to quantify the influence of both OH global burden and spatial distributions on top-down estimates of global, regional, and sectoral $CH_4$ emissions (the brown box with the solid line, Fig. 1). Because of the long lifetime of $CH_4$ relative to OH, the top-down estimates of regional $CH_4$ emissions can be influenced by both global total OH burden and OH spatial and seasonal distributions. To separate the influence of OH spatial distributions (including their seasonal variations) from that of the global annual mean [OH], we conduct Inv2, where all the prescribed OH fields are globally scaled to the global $[OH]_{GM-CH4}$ value of the INCA NMHC OH field in 2000 to get the same loss of $CH_4$ by OH (scaled OH fields). As such, Inv2 provides the uncertainty range of $CH_4$ emissions induced by OH spatial distribution in both horizontal and vertical directions as well as seasonal variations when assuming that the global total burden of OH can be precisely constrained (the brown box with the dashed line, Fig. 1). Thus, Inv1 (the inversions using original OH fields) and Inv2 (the inversions using scaled OH fields) yield upper (uncertainties from both global OH burden and spatial distributions) and lower (uncertainties only from OH spatial and seasonal distributions) limits of influences of OH on regional $CH_4$ emissions, respectively.

To quantify the influence of OH on $CH_4$ interannual emission changes, we also conduct Inv3 and Inv4 over 2007-2009, with 6 scaled OH fields (instead of 10 to limit computational time). While both of the inversions are done for 2007-2009 (Inv3 and Inv4), the OH variations during 2007-2009 (Inv3) and 2000-

2002 (Inv4) are used for the two inversions, respectively. Therefore, the difference Inv3－Inv2 reveal the impact of OH on $CH_4$ emission changes between the early and late 2000s (the yellow box with solid lines of Fig. 1), Inv3－Inv4 separates the impact of OH interannual variations, and the difference Inv4－Inv2 allows assessing the uncertainties of optimized $CH_4$ emission changes due to different OH spatial and seasonal distributions (the yellow boxes with dashed lines in Fig. 1).

Finally, we test the impact of OH year-to-year variations and trends on $CH_4$ emissions over 2001-2015 by running two long-term inversions (Inv5 and Inv6) with the OH fields simulated by CESM-WACCM only (the green box with dashed lines in Fig. 1). Inv5 is forced by the OH fields with both year-to-year variations and trends, while Inv6 is forced by the OH fields for the year 2000. For each group, only one experiment was done for computational reasons. We chose OH fields simulated by CESM1-WACCM because it shows the largest year-to-year OH variations and a positive trend of 0.35% $yr^{-1}$ during 2000-2010 among the CCMI OH fields (Zhao et al., 2019). Therefore, inversions using CESM1-WACCM OH are expected to yield an upper limit of the influence of OH variations on $CH_4$ emissions as seen from atmospheric chemistry models.

We evaluate the optimized $CH_4$ emissions by comparing the simulated $CH_4$ mixing ratios using prior and posterior $CH_4$ emissions with independent measurements from the NOAA/ESRL Aircraft Project. The location of the observation site (Table S1) and the vertical profile of the model bias in $CH_4$ mixing ratios compared with the aircraft observations (model minus observations) are shown in the supplement (Fig. S4a for Inv1 and Fig. S4b for Inv2). The comparisons with independent aircraft observations confirm the improvement of model-simulated $CH_4$ mixing ratios when using posterior emissions. All of the inversions in Inv1 and Inv2 reach small biases when compared with aircraft observations (right panel of Fig.S4a and Fig.S4b), which means that it is hard to distinguish which OH spatial and vertical distributions are more realistic in terms of quality of fit to these aircraft $CH_4$ observations. For Inv1, the root mean square errors

(RMSE $= \frac{\sqrt{\sum(model-observation)^2}}{n\_obs}$, n_obs is the number of observations) are reduced from up to more than 100ppbv (prior) emissions to ~10ppbv (posterior). For Inv2, although the $CH_4$ mixing ratios simulated using prior emissions already match well with aircraft observations (MSE=8-17ppbv), the posterior emissions still reduce the RMSE by up to 10ppbv.

In the following sections, to quantify uncertainties in top-down estimations of $CH_4$ emissions due to OH, we calculate OH-induced $CH_4$ emission differences and uncertainties as the standard deviation and the maximum minus mininimum values of the inversion results, respectively.

## 3 Results

## 3.1 The impacts of OH burden and spatial distributions on $CH_4$ emissions in 2000-2002

### 3.1.1 Global total $CH_4$ emissions

Based on the ensemble of the 10 different OH fields listed in Table 1, global total emissions inverted by our system in Inv1 vary from 518 to 757Tg $CH_4$ $yr^{-1}$ during the early 2000s (2000/07-2002/06). The highest $CH_4$ emissions exceeding 700Tg $yr^{-1}$ are calculated using MOCAGE and SOCOL3 OH fields, for which $[OH]_{GM-CH4}$ ($15.0\times10^5$ and $16.3\times10^5$ molec $cm^{-3}$) are much higher than those of other OH fields ($10.3-12.6\times10^5$ molec $cm^{-3}$), leading to a larger $CH_4$ sink, and as a consequence larger $CH_4$ emissions due to the mass balance constraint of atmospheric inversions. The high $[OH]_{GM-CH4}$ simulated by SOCOL3 and MOCAGE are mainly due to high surface and mid-tropospheric NO mixing ratio simulated by these two models (Zhao et al., 2019). As analyzed in Zhao et al. (2019), the lack of $N_2O_5$ heterogeneous hydrolysis (by both SOCOL3 and MOCAGE) and the overestimation of tropospheric NO production by $NO_2$ photolysis (by SOCOL3) are the major factors behind the overestimation of NO and OH.

The minimum-maximum range of the $CH_4$ emissions estimated by the 10 OH fields is almost similar to

the range estimated by previous bottom-up studies (542-852Tg yr[-1] given by Kirschke et al., 2013 and 583-861Tg yr[-1] given by Saunois et al, 2016) from GCP syntheses and much larger than that reported by an ensemble of top-down studies for 2000-2009 in Kirschke et al. (2013) (526-569Tg yr[-1]), Saunois et al. (2016) (535-566Tg yr[-1]) or the recent Saunois et al. (2019) (522-559 Tg yr[-1]). (Table 2 and Fig. 2). In the three top-down model ensembles, most of the inversion systems use TransCom OH fields, and the reported differences are mainly from different model transport and set-up of the inversion systems (e.g. the observations used in the inversions). Excluding the two outliers (MOCAGE and SOCOL-3) in Inv1, we find an uncertainty of about 17% in global methane emissions (518 to 611Tg yr[-1]) due to OH global burden and distributions, while transport model errors lead to only 5% of the uncertainty of the global methane budget (Table 3, Locatelli et al. (2013)). Our results indicate that considering different OH fields within top-down $CH_4$ inversions would lead to larger uncertainty on the top-down $CH_4$ budget.

Plotting top-down estimated $CH_4$ emissions against $[OH]_{GM-CH4}$, which directly reflects the global OH oxidizing efficiency with respect to $CH_4$ (Lawrence et al., 2001), reveals that the global total $CH_4$ emissions vary linearly with $[OH]_{GM-CH4}$ ($r^2$ >0.99, Fig. 2, right panel). The top-down estimation of global total $CH_4$ emissions ($EMIS_{CH4}$) can be approximately calculated as:

$$EMIS_{CH4}=40.4\times[OH]_{GM-CH4}+66.7 \quad\quad\quad (1)$$

Where a $1\times10^5$ molec cm[-3] (1%) increase in $[OH]_{GM-CH4}$ will increase the top-down estimated $CH_4$ emissions ($EMIS_{CH4}$) by 40.4 Tg yr[-1], consistent with that given by He et al. (2020) using full-chemistry modeling and a mass balance approach. Other $CH_4$ sinks including soil uptake and oxidation by $O^1(D)$, which are prescribed in this study, remove 66.7Tg yr[-1] $CH_4$. If uncertainties in all the $CH_4$ sinks were also considered, the correlation between optimized $CH_4$ emissions and the $[OH]_{GM-CH4}$ would be reduced. Using box-model inversions, previous studies calculated that a 4% ($0.4\times10^5$ molec cm[-3]) decrease in $[OH]_{GM}$ is equivalent to an increase of 22Tg yr[-1] $CH_4$ emissions (Rigby et al., 2017; Turner et al., 2017,

2019). If we apply the same $[OH]_{GM}$ changes in Eq.1 ($0.4\times10^5$molec cm$^{-3}$), the equivalent emissions change is 16Tg yr$^{-1}$, about 25% smaller than that given by Turner et al., (2017). This difference probably results from the different hemispheric mean reaction rates of OH+CH$_4$ applied in box models, but could also be due to different treatments of inter-hemispheric transport and stratospheric CH$_4$ loss in global 3D transport models compared to simplified box-models (Naus et al., 2019).

With the OH fields scaled to the same $[OH]_{GM-CH4}$ ($11.1\times10^5$molec cm$^{-3}$ ), the Inv2 simulations (assuming a global total OH burden well constrained) estimated global CH$_4$ emissions of $551\pm2$Tg yr$^{-1}$ (Table 3), as expected by the scaling. Differences in OH spatial distributions only lead to negligible uncertainty in global total CH$_4$ emissions estimated by top-down inversions.

### 3.1.2 Regional CH$_4$ emissions

**Inv1 and Inv2**

Since MOCAGE and SOCOL3 OH fields show much higher $[OH]_{GM}$ than constrained by MCF observations ($\sim10\times10^5$ molec cm$^{-3}$, e.g. Prinn et al., 2001; Bousquet et al., 2005) and give much higher estimates of CH$_4$ emissions (>700Tg yr$^{-1}$) than other OH fields, we exclude inversion results with these two OH fields from the following analyses.

In response to both global total OH burden and inter-hemispheric OH ratios (Table 1), CH$_4$ emissions over northern and southern hemispheres calculated by Inv1 (Table 2) vary from 368 to 424Tg yr$^{-1}$ ($401\pm$ 21Tg yr$^{-1}$) and 138 to 187Tg yr$^{-1}$ ($166\pm15$Tg yr$^{-1}$), respectively; resulting in a range in inter-hemispheric CH$_4$ emission difference (NH$-$SH) of 206-254Tg yr$^{-1}$ ($236\pm14$Tg yr$^{-1}$). When scaling all OH fields to the same loss for Inv2, the standard deviations of hemispheric CH$_4$ emissions are reduced to 7Tg CH$_4$ yr$^-$$^1$ for both hemispheres (Table 2), much smaller than those derived in Inv1 (21Tg yr$^{-1}$ and 15Tg yr$^{-1}$ over the northern and southern hemisphere, respectively). However, the CH$_4$ emission inter-hemispheric

difference calculated by Inv2 remains at $236 \pm 14$Tg yr$^{-1}$, similar to that calculated by Inv1, which indicates that the hemispheric CH$_4$ emissions differences are mainly determined by OH spatial distributions. Without the TransCom OH simulation, the inter-hemispheric CH$_4$ emission difference ranges between 232 and 246Tg yr$^{-1}$. The TransCom OH field, for which OH N/S ratio is 1.0, leads to an

inter-hemispheric CH$_4$ emission difference of 205Tg yr$^{-1}$, 35Tg yr$^{-1}$ (27Tg yr$^{-1}$) smaller than the mean (minimum) inter-hemispheric difference calculated using other OH fields (OH N/S ratio = 1.2-1.3). Previous studies show that differences in atmospheric transport models can lead to $\pm 28$Tg yr$^{-1}$ uncertainties in the top-down calculation of the inter-hemispheric CH$_4$ emission difference, using a single OH field – TransCom (Locatelli et al., 2013). Here, using a single atmospheric transport model, but

different OH fields, we find a $\pm 14$Tg yr$^{-1}$ uncertainty, about half of the atmospheric transport model uncertainty. Combining the two studies, one could expect more than 30Tg yr$^{-1}$ uncertainty in top-down estimates of the inter-hemispheric CH$_4$ emission difference, based on different atmospheric models and different OH fields.

Fig. 3 shows the optimized and prior CH$_4$ emissions calculated by Inv1 (top) and Inv2 (bottom) over latitudinal intervals (left panels) and large emitting regions (right panels). Compared with prior emissions, nearly all the optimized latitudinal and regional emissions show the same increment direction from prior emissions, but the magnitudes of the increment largely vary. The CH$_4$ emissions calculated by Inv1 amount to i) $147 \pm 14$Tg yr$^{-1}$ and are 1-47Tg yr$^{-1}$ higher than the prior estimate over the southern tropical

regions (30°S-0°), ii) $199 \pm 14$Tg yr$^{-1}$ and are 6-45Tg yr$^{-1}$ higher than the prior estimate over the northern tropical regions (0°-30°N), and iii) $174 \pm 8$Tg yr$^{-1}$ and are 1-26Tg yr$^{-1}$ lower than the prior estimate over the northern mid-latitude regions (30°N-60°N) (Table 3). The uncertainties in global OH burden and distributions lead to larger uncertainty (maximum$-$minimum) in top-down estimated CH$_4$ emissions over the tropics (>20% of multi-inversion mean) and smaller uncertainty over the northern mid-latitude

regions (14%) compare with that lead by transport model errors and different observations given by

Saunois et al. (2016) (13% over tropics and 20% over northern mid-latitude regions) (Table 3).

Over the large emitting regions Europe (EU), Canada (CAN), and China (CHN), optimized emissions are lower than the prior. The emissions calculated by Inv1 show the largest absolute OH induced differences over South America (SA, $73\pm9$Tg yr$^{-1}$), South Asia (SAS, $59\pm6$Tg yr$^{-1}$), and China ($42\pm5$Tg yr$^{-1}$) (Fig. 3, right panels and Table 3), of which the uncertainty (maximum$-$minimum) account for more than 20% of the multi-inversion mean emission over the corresponding regions (Table 3). Over high-latitude regions (Canada, Europe, and Russia), OH lead to small uncertainty ranges (<10Tg yr$^{-1}$). At the model grid-scale, the uncertainty range due to OH can be much larger than the regional mean (middle panel of Fig. 3), for example, larger than 50% of the multi-inversion mean emissions over South America and East Asia. As shown in Table 3, at regional scales, the uncertainty (maximum$-$minimum) in top-down estimated CH$_4$ emissions due to different OH global burden and distributions over Asia and South America (~37% of multi-inversion mean) are of the same order than those lead by transport errors (25% and 48%) or given by Saunois et al. (2016) (~40%). Over other regions, using different OH fields lead to smaller uncertainties (11%-18%) compared to other causes of errors (23%-70%) (Table 3).

The uncertainties in the top-down estimated regional emissions are not only due to inter-model differences of the regional OH fields but also rely on the distribution of the surface observations used in the inversions. Over the regions with large prior emissions but less constrained by observations (e.g. South America, South Asia, and China), our OH analysis leads to larger uncertainties than regions that are well constrained by observations (e.g. the North America and Canada) (Fig. S3). The results may indicate that on the regional scale, the top-down estimated CH$_4$ emissions and the uncertainties lead by OH are specific to the observation system retained. If more surface observations (e.g. in the southern hemisphere) or satellite columns with a more even global coverage were included in our inversions, spatial patterns of the top-down estimated CH$_4$ emissions and their uncertainties (as shown by Fig.3) could be different.

**Comparing Inv1 and Inv2**

We now compare the inversion results using the original OH fields (Inv1) with those using scaled OH fields (Inv2) to estimate how much the optimized regional $CH_4$ emission differences of Inv1 are dominated by OH spatial and seasonal distributions versus the global OH burden. Applying one single global scaling factor per model reduces the inter-model differences of original OH fields by 33%, 67%, and 33% in the southern tropics(0°-30°S), northern tropics(0°-30°N), and northern mid and high latitudes (30°-90°N) (Table S2). This scaling results in 57%, 93%, and 22% reduction of OH induced latitudinal $CH_4$ emission differences, respectively for the southern tropics(0°-30°S), northern tropics(0°-30°N), and northern mid and high latitudes (30°-90°N) (Fig. 3, left panels comparing standard deviations of Inv1 and Inv2). At the regional scale (Fig. 3, right panel and Table 3), the OH spatial distribution-induced $CH_4$ emission differences (standard deviation of Inv2) account for 50% of the differences due to both OH burden and spatial distributions (standard deviation of Inv1) over northern mid-latitude regions (China, North America) and South America. Over northern tropical regions (Southern Asia and Southeast Asia), the OH spatial distribution induces negligible $CH_4$ emission differences.

The comparison of Inv1 and Inv2 reveals that methane emissions in tropical regions are less sensitive to OH spatial distribution than mid- and high-latitude regions in our framework. One possible explanation could be the location of monitoring sites. Over tropical regions, $CH_4$ emissions are less constrained (with few to none observation sites near source regions) than in the northern extra-tropics, where several monitoring sites located at or near the regions with high $CH_4$ emission rates and high OH uncertainties (e.g. North America, Europe, and downwind of East Asia). Thus, $CH_4$ emissions over the tropical regions mainly contribute to match the global total $CH_4$ sinks (instead of the sinks over the tropical regions only) estimated by inversion systems. When all OH fields are scaled to the same $CH_4$ losses (Inv2), differences of emissions over the tropical regions are therefore largely reduced.

### 3.1.3 Global and regional CH$_4$ emissions per source category

Fig. 4 compares optimized and prior global total CH$_4$ emissions and the difference between the prior and optimized CH$_4$ emissions for four broad source categories: wetlands, agriculture and waste (named Agri-waste), fossil fuels, and other natural sources. We attribute the optimized emissions to different source sectors depending on the relative strength of individual prior sources in each grid-cell. With original OH fields, Inv1 calculates CH$_4$ emissions of $203\pm15$Tg yr$^{-1}$ for wetlands, $209\pm12$Tg yr$^{-1}$ for Agri-waste, $89\pm4$Tg yr$^{-1}$ for fossil fuel, and $66\pm3$Tg yr$^{-1}$ for other natural sources. Optimized emissions of the four sectors are $23\pm15$Tgyr$^{-1}$ (-2-42Tg yr$^{-1}$ ), $13\pm12$Tg yr$^{-1}$ (-3-29Tg yr$^{-1}$), $5\pm4$Tg yr$^{-1}$ (-1-9Tg yr$^{-1}$), and $4\pm3$Tg yr$^{-1}$ (0.1-8Tg yr$^{-1}$) higher than the prior emissions, respectively. Although Inv2 is conducted with scaled OH fields and all inversions calculate similar global total CH$_4$ emissions ($551\pm2$Tg yr$^{-1}$), optimized CH$_4$ emissions still show some uncertainties due to OH (as standard deviation) (3Tg yr$^{-1}$ for wetland emissions and 2Tg yr$^{-1}$ for agriculture and waste, yellow boxplots in Fig. 4), in response to different OH spatial distributions.

We have further calculated CH$_4$ emissions per source category and per region estimated by Inv1 (Fig. 5), to explore the contribution of each region to the OH-induced sectoral emission uncertainties. Wetland CH$_4$ emissions mainly dominate emissions over Northern South America, Africa, South and East Asia, and Canada. Northern South America ($53\pm7$Tg yr$^{-1}$) and Africa ($30\pm2$Tg yr$^{-1}$) contribute most of the global total OH induced wetland emission differences and are 1-22Tg yr$^{-1}$ and 1-8Tg yr$^{-1}$ higher than prior emissions, respectively. In contrast to the higher wetland emissions than prior ones over tropical regions, optimized boreal wetland emissions (in Canada) are 6-9Tg yr$^{-1}$ lower than prior emissions, consistent with lower top-down estimations than the prior given by Saunois et al. (2016). Agriculture and waste emissions are most intensive over China ($25\pm3$Tg yr$^{-1}$) and South Asia (SAS) ($39\pm3$Tg yr$^{-1}$). The optimized inventories show lower agriculture and waste emissions over China (0.6-10Tg yr$^{-1}$) and Europe (1-3Tg

yr$^{-1}$) and much higher emissions over SAS (4-13Tg yr$^{-1}$) compared with the prior emission inventory. The OH induced differences in fossil fuel emissions are found mainly in China and Africa, which are 0.8-5Tg yr$^{-1}$ lower and 0.6-3Tg higher than prior emissions, respectively. In agreement with the previous regional discussion, scaling the OH (Inv2) highly reduces the uncertainties attributable to different OH over the tropical regions but not for the mid-high latitude regions. In Inv2, the largest CH$_4$ emission differences due to different OH spatial distribution are found for wetland emissions in South America (60$\pm$4Tg yr$^{-1}$), agriculture and waste emissions in South Asia (17$\pm$1Tg yr$^{-1}$) and China (24$\pm$2Tg yr$^{-1}$), and fossil fuel emissions in China (8$\pm$0.7Tg yr$^{-1}$) and Russia (9$\pm$0.4Tg yr$^{-1}$).

Previous studies have highlighted that anthropogenic emissions over China are largely overestimated by bottom-up emissions inventories compared with top-down estimates (Kirschke et al., 2013; Tohjima et al., 2014; Saunois et al., 2016). In our study, total anthropogenic emissions (agriculture, waste, and fossil fuel) over China are 1-15Tg yr$^{-1}$ lower than the prior bottom-up inventory as calculated by Inv1, and 7-14Tg yr$^{-1}$ as calculated by Inv2, with the lowest emissions calculated with the TransCom OH field (for both Inv1 and Inv2). The TransCom OH field is the one most widely used in current top-down CH$_4$ emission estimations but shows much lower [OH] over China than other OH fields (Zhao et al., 2019), which may be due to the use of the same NO$_x$ profile over East Asia as for remote regions based on the observations of 1990s when constructing the TransCom OH field (Spivakovsky et al., 2000). Thus, the large reduction of top-down estimated anthropogenic CH$_4$ emissions over China as compared to the prior emissions may be partly due to an underestimation of [OH] over China in the TransCom field.

## 3.2 Impact of OH on CH$_4$ emission changes between 2000-2002 and 2007-2009

As shown in Table 4, the global mean [OH] simulated by CCMI models increased by 0.7%-1.8% from 2000-2002 to 2007-2009, in response to anthropogenic emissions and climate change (Zhao et al., 2019), whereas the INCA-NMHC model-simulated global [OH] shows a slight decrease of 0.5%. The TransCom

OH field, being constant over time, shows no change. The increase in global mean [OH] mainly results from the combination of a higher increase in the tropics compared to the northern extra-tropics and a slight decrease in the southern extra-tropics. As a result, the changes in OH between the two periods show different patterns between regions. We have conducted inversions for 2007-2009 with scaled OH fields (Inv3) to explore how uncertainties in OH (both spatial and seasonal distribution and interannual changes) can influence the top-down estimates of temporal $CH_4$ emission changes from the early 2000s (2000/07-2002/06, Inv2) to the late 2000s (2007/07-2009/06, Inv3) (Inv3－Inv2). We have also performed Inv4 for 2007-2009 but using OH fields of 2000-2002 to separate the contribution of OH from different time periods (Inv3－Inv4).

### 3.2.1 Global total emission changes between 2000-2002 and 2007-2009

**Total emission changes.** From the early 2000s (Inv2) to the late 2000s (Inv3), the top-down estimated $CH_4$ emissions increased by 21.9±5.7Tg yr$^{-1}$ (16.6-30.0Tg yr$^{-1}$, Table 5). The largest $CH_4$ increase of 30.0Tg yr$^{-1}$ is estimated with CESM1-WACCM OH fields (for which OH increased by 1.8% from 2000-2002 to 2007-2009), 13.4Tg yr$^{-1}$ higher than the smallest increase of 16.6 yr$^{-1}$ estimated with the INCA NMHC OH field (for which OH decreased by 0.5% from 2000-2002 to 2007-2009). In Saunois et al. (2017), the minimum-maximum uncertainty range of emission changes between 2002-2006 and 2008-2012 was 16Tg yr$^{-1}$. This means that the uncertainty attributable to uncertainty in OH fields (13.4Tg yr$^{-1}$), is comparable to the minimum-maximum uncertainty resulting from using different atmospheric chemistry transport models and observations (surface and satellite), but mostly constant OH over time (16Tg yr$^{-1}$, Saunois et al., 2017).

**Spatial versus temporal OH effects.** Only changing OH from 2000-2002 (Inv4) to 2007-2009 (Inv3), top-down estimated $CH_4$ emissions due to OH interannual changes are +5.1±6.4Tg yr$^{-1}$ (-2.7-13.5Tg yr$^{-1}$

[1], Table 5), which contribute 25% of total optimized emission changes (Inv3－inv2) between the early and late 2000s (21.9±5.7Tg yr$^{-1}$, Table 5). As listed in Table 5, the largest emission increase due to OH interannual changes are calculated using MRI-ESM1r1 OH fields, for which a 1.1% global increase in OH can up to double the top-down estimation of $CH_4$ emission increase from the early to the late 2000s. This result indicates that a large bias likely exists in the former top-down estimation of the $CH_4$ emission

trend calculated without considering OH changes (Saunois et al., 2017).

Keeping OH fields from 2000-2002, top-down estimated $CH_4$ emissions increase by 16.9±1.9Tg yr$^{-1}$ (14.3-19.3Tg yr$^{-1}$, Table 5) between the early 2000s (Inv2) to the late 2000s (Inv4) in response to increasing atmospheric $CH_4$ mixing ratios and temperature. This represents 75% of total optimized

emission changes (Inv3－inv2) between the early and late 2000s (21.9±5.7Tg yr$^{-1}$, Table 5). The 1.9Tg yr$^{-1}$ uncertainty (as standard deviation) is due to the different OH spatial and seasonal distributions, indicating that OH spatial and seasonal distributions, which are not considered in box-models, can also contribute to the uncertainties in optimized $CH_4$ emission changes.

**3.2.2 Emission changes by source types and regions**

**Total emission changes.** We further analyze the influence of OH (both spatial distributions and interannual variations) on the top-down estimated sectoral and regional $CH_4$ emission changes from the early 2000s (Inv2) to the late 2000s (Inv3). As shown in Fig. 6 (top panels), the smaller increase of the optimized global $CH_4$ emissions from the early 2000s to the late 2000s (21.9±5.7Tg yr$^{-1}$) compared to the

prior change (39.4Tg yr$^{-1}$) is mainly due to decrease in wetland emissions over the southern tropics (-4.4±1.5Tg yr$^{-1}$, 15 °S-0 °) and northern mid-latitude regions (-3.4±0.4Tg yr$^{-1}$, 45 °N-60 °N) in contrast to climatology prior wetland emissions, and a lower fossil fuel emission increase over 30 °N-45 °N (3.7± 0.4Tg yr$^{-1}$) compared to prior emission increase (8.9Tg yr$^{-1}$). Wetlands (-3.5±2.5Tg yr$^{-1}$) and agriculture and waste (14.2±2.1Tg yr$^{-1}$) contribute most of the total OH induced uncertainty in global total emission

changes ($21.9\pm5.7$Tg yr$^{-1}$) from the early 2000s (Inv2) to the late 2000s (Inv3) (Table 6), whereas fossil fuel emissions ($8.7\pm0.8$Tg yr$^{-1}$) show smaller uncertainty.

Considering emissions over latitudinal bands (Fig. 6, top panels), the largest spread of emission changes are found over the southern tropics (15 °S –tropics, -1.3 to -6.5Tg yr$^{-1}$), northern subtropics (15 °N-30 °N,

15.5 to 20.0Tg yr$^{-1}$), and northern extratropical regions (30 °N-45 °N, 7.6 to 10.7Tg yr$^{-1}$). The spread over the southern tropics is dominated by emission changes from wetlands (-2.3 to -6.3Tg yr$^{-1}$), over northern subtropics by agriculture and waste (7.3 to 10.0Tg yr$^{-1}$), and over northern extratropical regions by agriculture and waste (3.7 to 5.1Tg yr$^{-1}$) and fossil fuels (3.3 to 4.3Tg yr$^{-1}$). At the regional scale (Fig. 7, top panels), northern South America (-1.2 to -5.2Tg yr$^{-1}$), South Asia (9.1 to 12.4Tg yr$^{-1}$), and China (-

0.1 to 4.9Tg yr$^{-1}$) show the largest differences in emission changes from the early 2000s to the late 2000s. The multi-inversions calculated emission changes in China disagree in sign (Fig. 7; top panels), mainly due to differences in the agriculture and waste sector, which range from 1.3Tg yr$^{-1}$ decrease to 1.5Tg yr$^{-1}$ increase from the early 2000s to the late 2000s.

We now compare the uncertainty of top-down estimated CH$_4$ emission changes from the early to the late 2000s due to different OH spatial-temporal variations with that ensemble of top-down studies given by Saunois et al. (2017). For the sectoral emissions, the emission changes from agriculture and waste and from wetland show the largest uncertainties (more than 50% of multi-inversions mean, Inv3－Inv2 in Table 6) induced by OH spatial-temporal variations, comparable to that given by Saunois et al. (2017).

On the contrary, the uncertainty of fossil fuel emission changes (24% of multi-inversions mean) is much smaller than that given by Saunois et al. (2017). For regional CH$_4$ emission changes, the uncertainty induced by OH spatial-temporal variations is usually larger than the multi-inversion mean emission changes (except South Asia) and similar to that given by Saunois et al. (2017). The large differences existing in different top-down estimated regional and sectoral emission changes are mainly attributed to

model transport errors in Saunois et al. (2017). Here, our results show that uncertainties due to OH spatio-temporal variations can lead to similar biases in top-down estimated $CH_4$ emission changes.

**Spatial versus temporal effects.** We now separate influences of OH interannual changes (Inv3－Inv4) on optimized regional $CH_4$ emission changes. As shown in the bottom panels of Fig. 6 and Fig. 7, at the
regional scale, OH interannual changes mainly perturb top-down estimated $CH_4$ emission changes (Fig. 6; bottom panel) over the southern tropics (0°-15°S, -3.4-5.9Tg $yr^{-1}$) and northern subtropics (15°N-30°N, 0-5.4Tg $yr^{-1}$). This corresponds to the two largest spreads observed in Fig.7 (bottom panel) associated with wetland emissions over northern South America (-2.0-3.5Tg $yr^{-1}$) and with agriculture and waste emissions over South Asia (-0.2-2.6Tg $yr^{-1}$). Among the four emission sectors, wetland emissions (mainly
southern tropical wetland) show the largest increase (3.3-4.8Tg $yr^{-1}$) in response to OH temporal changes (Table 6), which account for 60% of total wetland emission changes between these two periods.

The OH spatial and seasonal distribution can lead to large uncertainties in regional $CH_4$ emission changes. For regional and latitudinal scales, the spreads (uncertainty ranges) of Inv4－inv2 (OH fixed to 2000-
2002) (Fig. S5 and Fig. S6) are comparable to the spread of regional and latitudinal emission changes lead by both OH interannual changes and spatial and seasonal distributions (Inv3－Inv2) (top panels of Fig. 6 and Fig. 7) as mentioned above (e.g., 2.6-5.4Tg $yr^{-1}$ decrease over northern South America, a 6.1-11.1Tg $yr^{-1}$ increase over South Asia, and a 0.4-4.2Tg $yr^{-1}$ increase over China). These results show that even if the global total OH burden is well constrained (as in Inv4 and Inv2, where all OH fields are scaled
to the same $[OH]_{GM-CH4,}$ and the differences in optimized $CH_4$ emissions changes from the early 2000s to the late 2000s are only due to different OH spatial distributions), top-down estimates of sectoral and regional temporal $CH_4$ emission changes remain highly uncertain.

**3.3 Impacts of OH on year to year variations of $CH_4$ emissions from 2001 to 2015**

To infer the influence of OH year-to-year variations on top-down optimized long-term $CH_4$ emission changes, we conducted two inversions, Inv5 and Inv6. Inv5 calculates optimized $CH_4$ emissions from 2001-2015 with the CESM1-WACCM OH field varying from one year to the next, while Inv6 uses the CESM1-WACCM OH field but fixed to the year 2000. The choice of the CESM1-WACCM OH field is explained in Sect. 2.3 above. As shown in Fig. 8, the $[OH]_{GM-CH4}$ of the CESM1-WACCM OH field

increases by $0.47 \times 10^5$ molec cm$^{-3}$ (4.2%) from 2001 to 2011 and then decreases by $0.13 \times 10^5$ molec cm$^{-3}$ (1.1%) from 2011 to 2015.

With OH fixed to the year 2000 (Inv6), global $CH_4$ emissions stall at $550 \pm 2$Tg yr$^{-1}$ during 2001-2003, decrease to 538Tg yr$^{-1}$ in 2004, which is different from the continuous increase of $CH_4$ emissions given

by the bottom-up inventory (Fig. 8, top panel). After 2004, global total $CH_4$ emissions show a positive trend of $3.5 \pm 1.8$Tg yr$^{-2}$ (P<0.05), but smaller than the prior bottom-up inventory ($4.3 \pm 0.6$Tg yr$^{-2}$ (P<0.05)). Both stalled/decreased emissions during 2001-2004 and increasing trend after 2004 are consistent with previous top-down estimations (Saunois et al., 2017).

The trend of global $CH_4$ emission during 2004-2016 calculated by Inv5 (using varying OH) is $4.8 \pm 1.8$Tg yr$^{-2}$ (P<0.05), which is 1.3Tg yr$^{-2}$ (36%) higher than that calculated by Inv6 (OH fixed to 2000) due to the small increase in [OH], and also 0.5Tg yr$^{-2}$ higher than the prior emission trend ($4.3 \pm 0.6$Tg yr$^{-2}$). Accounting for the OH increasing trend leads to increasing the prior trend in Inv5 instead of decreasing it in Inv6. When calculating the differences between Inv5 and Inv6 for different latitude intervals, we find

that before 2004, differences between Inv5 and Inv6 are mainly contributed by northern middle-latitude regions, whereas after 2004 they are dominated by tropical regions (Fig. 8, bottom).

We further compare $CH_4$ emission trends for the four previously defined emission sectors and the ten continental regions between Inv5 and Inv6. As shown in Fig. 9, the positive global $CH_4$ emission trend

during 2004-2016 is mainly contributed by anthropogenic sources from agriculture and waste, and fossil fuel-related activities, which are $1.9\pm0.7$Tg yr$^{-2}$, and $2.3\pm0.4$Tg yr$^{-2}$, respectively, as calculated by Inv6 (fixed OH). Wetland emissions show a small negative trend ($-0.5\pm0.7$Tg yr$^{-2}$) and other natural emissions do not show a significant trend ($0.04\pm0.6$Tg yr$^{-2}$). Both sectors show large uncertainties in their trends reflecting large year-to-year variations. When considering [OH] variations, Inv5 estimates a higher agriculture and waste emission trend ($2.4\pm0.8$Tg yr$^{-2}$) compared to Inv6, mainly contributed by China ($1.5\pm0.5$Tg yr$^{-2}$ for Inv6 versus $1.7\pm0.5$Tg yr$^{-2}$ for Inv5) and southern South America ($-0.1\pm0.1$Tg yr$^{-2}$ for Inv6 versus $0.1\pm0.3$Tg yr$^{-2}$ for Inv5). Accounting for interannual OH variations the negative wetland emission trend reduces to near zero ($0.1\pm0.6$Tg yr$^{-2}$), mainly due to increased emission trends over northern South America ($-0.3\pm0.3$Tg yr$^{-2}$ for Inv6 versus $0.2\pm0.5$Tg yr$^{-2}$ for Inv5). In contrast to agriculture and waste, and wetland emissions, fossil fuel emissions have a similar positive trend of $2.4\pm0.4$Tg yr$^{-2}$ in Inv5 and Inv6. This result comes from a higher $CH_4$ emission trend over China calculated by Inv5 balanced by a lower $CH_4$ emission trend over America and Russia ($0.2\pm0.2$Tg yr$^{-2}$ for Inv6 versus $0.1\pm0.3$Tg yr$^{-2}$ for Inv5) since the CESM1-WACCM OH field shows a significant negative [OH] trend over America (Zhao et al., 2019).

## 4 Conclusions and discussion

In this study, we have performed six groups of variational Bayesian inversions (top-down, 34 inversions in total) using up to 10 different prescribed OH fields to quantify the influence of OH burden, interannual variations, and spatial and seasonal distributions on the top-down estimation of i) global total, regional, and sectoral $CH_4$ emissions, ii) emission changes between the early 2000s and late 2000s, and iii) year-to-year emission variations. Our top-down system estimates monthly $CH_4$ emissions by assimilating surface observations with atmospheric transport of $CH_4$ calculated by the offline version LMDz5B of the LMDz atmospheric model using different prescribed OH fields.

Based on the ensemble of 10 original OH fields ($[OH]_{GM-CH4}$: 10.3-16.3 $\times10^5$ molec cm$^{-3}$), the global total CH$_4$ emissions inverted by our system vary from 518 to 757Tg yr$^{-1}$ during the early 2000s, similar to the CH$_4$ emission range estimated by previous bottom-up syntheses and larger than the range reported by the top-down studies (Kirschke et al., 2013; Saunois et al, 2016;2019). The top-down estimated global total CH$_4$ emission varies linearly with $[OH]_{GM-CH4}$, which indicates that at the global scale, a small uncertainty of $1\times10^5$ molec cm$^{-3}$ (10%) $[OH]_{GM-CH4}$ can result in 40.4Tg yr$^{-1}$ uncertainties in optimized CH$_4$ emissions.

At regional scale (excluding the two highest OH fields), CH$_4$ emission uncertainties due to different OH global burdens and distributions are largest over South America (37% of multi-inversion mean), South Asia (24%), and China (39%), resulting in significant uncertainties in optimized emissions from the wetland and agriculture and waste sectors. These uncertainties are comparable in these regions with those due to model transport errors and inversion system set-up (Locatelli et al., 2013; Saunois et al., 2016). For these regions, the uncertainty due to OH is critical for understanding their methane budget. In other regions, OH leads to smaller uncertainties compared to that given by Locatelli et al. (2013) and Saunois et al. (2016). By performing inversions with globally-scaled OH fields, we calculated that emission uncertainties due to different OH spatial and seasonal distributions account for ~50% of total uncertainties (induced by both different OH burden and different OH spatial and seasonal distributions) over mid-high latitude regions and South America. CH$_4$ emission differences due to OH spatial distributions are the largest in northern South America and China but are negligible over South Asia and other northern tropical regions. Based on CH$_4$ emission optimization with surface observations, our study shows that tropical regions appear more sensitive to OH global burden (as less constrained regions used to achieve the global mass balance of the methane budget) and mid-to-high latitude regions are found sensitive to both OH global burden and spatial distributions.

The global CH$_4$ emission change between 2000-2002 and 2007-2009 as estimated by top-down inversions

using 6 different OH fields, is $21.9\pm5.7$Tg yr$^{-1}$, of which 25% ($5.1\pm6.4$Tg yr$^{-1}$) is contributed by OH interannual variations (mainly by an increase in [OH]), while 75% can be attributed to emission changes resulting from the increase in observed CH$_4$ mixing ratios and atmospheric temperature (considering constant OH). Among the four emission sectors, wetland emissions (mainly southern tropical wetlands) show the largest increase of $2.1\pm3.4$Tg yr$^{-1}$ in response to OH temporal changes, which account for 60% of total wetland emission changes between these two periods. For global total emission changes, OH spatial distributions lead to lower uncertainties than interannual variations (1.9Tg yr$^{-1}$ versus 6.4Tg yr$^{-1}$), but at the regional scale, OH spatial distributions and interannual variations are of equal importance for quantifying CH$_4$ emission changes.

As the modeled OH used here mainly shows an increase in [OH] (meaning increasing CH$_4$ sink) during the 2000s, our inversion using year-to-year OH variations infers a 36% higher CH$_4$ emission trend compared with an inversion driven by climatological OH over the 2001-2015 period. The different OH fields from CCMI models consistently show increasing OH trends during 2000-2010 (Zhao et al., 2019). These variations disagree with MCF-constrained [OH], which show a decrease of $8\pm11$% during 2004-2014 and 7% during 2003-2016 estimated by Rigby et al. (2017) and Turner et al. (2017), respectively. A drop of OH between 2006-2007 (Rigby et al., 2008, Bousquet et al., 2011) is captured by CESM1-WACCM OH fields but with (possibly) smaller changes (1%) compared to (the very uncertain) $4\pm14$% changes constraint by MCF (Rigby et al., 2008). This OH drop in 2006-2007 results in a 6Tg yr$^{-1}$ smaller increase of CH$_4$ emissions between 2006 and 2007 compared to that derived using constant OH. However, such [OH] drop is treated as a year-to-year variation instead of a trend, and cannot fully explain the resumption of CH$_4$ growth from 2006-2007. Thus, during 2004-2010, at the decadal timescale, if the CCMI models represent the OH trend properly, a higher increasing trend of CH$_4$ emissions is needed to match the CH$_4$ observations (compared to the CH$_4$ emission trend derived using constant OH). After 2010, CCMI models simulated OH trends of different signs (Zhao et al., 2019), thus the influence on the CH$_4$

emission trend is more uncertain.

The trend and interannual variations of tropospheric OH burden are determined by both precursor emissions from anthropogenic and natural sources and climate change (Holmes et al., 2013; Murray et al., 2014). Based on satellite observations, Gauber et al. (2017) estimated that ~20% decrease in atmospheric

CO concentrations during 2002-2013 led to an ~8% increase in atmospheric [OH]. The El Niño-Southern Oscillation (ENSO) has been proven to impact the tropospheric OH burden and $CH_4$ lifetime mainly through changes in biomass burning from CO (Nicely et al., 2020; Nguyen et al, 2020) and in NO emission from lightning (Murrary et al., 2013; Turner et al., 2018). The ENSO signal is weak during the early 2000s, resulting in small interannual variations of tropospheric OH burden (Zhao et al., 2019). The

mechanisms of OH variations related to ENSO and their impacts on the $CH_4$ budget need to be explored by inversions, but over a longer time period than this study (e.g. 1980-2010, Zhao et al., 2020).

Compared to previous box-model studies (e.g. Rigby et al., 2017; Turner et al. 2017), the inversions performed in this study take advantage of 4D OH fields from CCMI to quantify impacts on regional and

sectoral emission estimations. Our results indicate that OH spatial distributions, which are difficult to obtain from proxy observations (e.g. MCF), are equally important as the global OH burden for constraining $CH_4$ emissions over mid- and high-latitude regions. Constraining global annual mean OH based on proxy observations (e.g. Zhang et al., 2018; Maasakkers et al., 2019) provides a constraint on global total methane emissions, through the necessity of balancing the global budget (sum of source −

sum of sinks = atmospheric growth rate). It also largely reduces uncertainties in optimized $CH_4$ emissions due to OH over most of the tropical regions but not over South America and overall mid-high latitude regions. Also, the spatial and seasonal distributions of OH is found critical to properly infer temporal changes of regional and sectoral $CH_4$ emissions.

Top-down inversions, particularly variational Bayesian systems, are powerful tools to infer greenhouse gas budgets, in particular, methane the target of this study. However, they suffer from some limitations impacting the budget uncertainty. Some work has been done regarding atmospheric transport errors (e.g., Locatelli et al., 2013, 2015) and sensitivity to observation constraints (Locatelli et al., 2015; Houweling et al., 2000), but less on the chemistry side of the budget. Overall, our study significantly contributes to

assessing the impact of OH uncertainty on the $CH_4$ budget. We have shown that it is insufficient to consider a unique OH field, constant over time, to fully understand and assess the global $CH_4$ budget and its changes over time. Indeed, further work is needed to help determining OH fields to be used in future variational top-down inversion studies to properly account for changes in both source and sinks. There are different ways to optimize the current OH fields. One way can be to build semi-empirical OH fields

by combining atmospheric chemistry models and observation-based meteorological data and chemical species concentrations (e.g. $NO_x$, CO, VOCs. etc) as initiated in Spivakovsky et al. (2000); another way is conducting multi-species variational inversion of OH (e.g. Zheng et al., 2019) with such HFC species (Liang et al., 2017), formaldehyde (Glenn et al., 2019), $CH_4$ (Zhang et al., 2018; Maasakkers et al., 2019), or CO (Zheng et al., 2019). In addition, as suggested by Prather et al. (2017), the OH inversion would

benefit from including in their prior data the responses of [OH] to variations of the precursor emissions (e.g. biomass burning and lighting) using the uncertainties estimated by 3D models. These resulting OH fields should include a mean 3D global [OH] distribution, associated with temporal variations and uncertainties. A lot remains to be done to better constrain the chemistry side of the global methane budget, a critical step toward its closure.


**Data availability**
The CCMI OH fields are available at the Centre for Environmental Data Analysis (CEDA; http://data.ceda.ac.uk/badc/wcrp-ccmi/data/CCMI-1/output; Hegglin and Lamarque, 2015), the Natural Environment Research Council's Data Repository for Atmospheric Science and Earth
Observation.    The    CESM1-WACCM    outputs    for    CCMI    are    available

at http://www.earthsystemgrid.org (Climate Data Gateway at NCAR, 2019). The surface observations for $CH_4$ inversions are available at the World Data Centre for Greenhouse Gases (WDCGG, https://gaw.kishou.go.jp/, 2019). Other datasets, including INCA OH fields and optimized $CH_4$ emissions, can be accessed by contacting the corresponding author.

**Author contributions**

YZ, MS, and PB designed the inversion experiments, analyzed results, and wrote the manuscript. BZ and XL helped with data preparation and inversion setup. JC and RJ discussed the results. AB developed the LMDz code for running $CH_4$ inversions. MH provided the CCMI OH fields. DW, ED, MR, and RL provided the atmospheric in situ data. All coauthors commented on the paper.

**Acknowledgements**

This work benefited from the expertise of the Global Carbon Project methane initiative.

We acknowledge the modeling groups for making their simulations available for this analysis, the joint WCRP SPARC/IGAC Chemistry–Climate Model Initiative (CCMI) for organizing and coordinating the model simulations and data analysis activity, and the British Atmospheric Data Centre (BADC) for collecting and archiving the CCMI model output.

**Financial support**

This research has been supported by the Gordon and Betty Moore Foundation (grant no. GBMF5439), "Advancing Understanding of the Global Methane Cycle".

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

## Tables

**Table 1.** Global tropospheric mean [OH] ($\times 10^5$ molec cm$^{-3}$) and inter-hemispheric OH ratios (N/S) averaged over 2000-2002 for 10 OH fields used in this study. The global tropospheric [OH] weighted by reaction with CH$_4$ ([OH]$_{GM-CH4}$) and weighted by dry air-mass ([OH]$_{GM-M}$) are both given.

|  | [OH]$_{GM-CH4}$ | [OH]$_{GM-M}$ | N/S |
|---|---|---|---|
| TransCom | 10.6 | 10.0 | 1.0 |
| INCA NMHC-AER-S | 10.3 | 9.4 | 1.3 |
| INCA NMHC | 11.1 | 10.4 | 1.2 |
| CESM1-WACCM | 11.9 | 11.4 | 1.3 |
| CMAM | 12.2 | 11.3 | 1.2 |
| EMAC-L90MA | 11.8 | 11.5 | 1.2 |
| GEOSCCM | 12.6 | 12.3 | 1.2 |
| MOCAGE | 15.0 | 12.5 | 1.5 |
| MRI-ESM1r1 | 10.9 | 10.6 | 1.2 |
| SOCOL3 | 16.3 | 14.4 | 1.5 |
| Mean±SD | 12.3±3.8 | 11.4±2.8 | 1.3±0.3 |
| Mean±SD (8 OH)[1] | 11.3±0.8 | 10.9±0.9 | 1.2±0.1 |

[1] The OH fields simulated by SOCOL3 and MOCAGE are excluded.

**Table 2.** The global total, hemispheric CH$_4$ emissions, and inter-hemispheric difference of CH$_4$ emissions calculated by Inv1 and Inv2 during the early 2000s (2000/07/01-2002/06/01) in Tg yr$^{-1}$.

| Unit: Tg yr$^{-1}$ | Inv1 original OH | | | | Inv2 scaled OH | | | |
|---|---|---|---|---|---|---|---|---|
|  | **Global** | **0-90°N** | **90°S-0** | **N-S$_{Inv1}$** | **Global** | **0-90°N** | **90°S-0** | **N-S$_{Inv2}$** |
| **Prior** | 522 | 384 | 138 | 246 | 522 | 384 | 138 | 246 |
| **TransCom** | 530 | 368 | 162 | 206 | 549 | 377 | 172 | 205 |
| **INCA NMHC-AER-S** | 518 | 380 | 138 | 242 | 553 | 399 | 154 | 245 |
| **INCA NMHC** | 552 | 392 | 160 | 232 | 552 | 392 | 160 | 232 |
| **CESM1-WACCM** | 587 | 420 | 166 | 254 | 551 | 400 | 151 | 249 |
| **CMAM** | 599 | 419 | 180 | 239 | 553 | 399 | 154 | 245 |
| **EMAC-L90MA** | 589 | 414 | 175 | 239 | 555 | 396 | 159 | 237 |
| **GEOSCCM** | 611 | 424 | 187 | 237 | 550 | 392 | 159 | 233 |

| | | | | | | | |
|---|---|---|---|---|---|---|---|
| **MOCAGE** | 716 | /[a] | / | / | / | / | / | / |
| **MRI-ESM1r1** | 553 | 396 | 156 | 240 | 548 | 396 | 152 | 244 |
| **SOCOL3** | 757 | / | / | / | / | / | / | / |
| **Mean±SD** | 601±78 | 401±21 | 166±15 | 236±14 | 551±2 | 393±7 | 158±7 | 236±14 |

[a] We do not analyze the hemispheric $CH_4$ emission estimated with MOCAGE and SOCOL3 OH field since inversions using the two OH fields calculate much higher $CH_4$ emissions than using other OH fields.

**Table 3.** Global, latitudinal, and regional $CH_4$ emission in Tg yr$^{-1}$ (mean±SD and the [min-max] range of
the inversions) calculated by Inv1 and Inv2 during the early 2000s (2000/07/01-2002/06/01) in Tg yr$^{-1}$ (excluding MOCAGE and SOCOL-3). The uncertainties (Unc.= (max−min)/multi-inversions mean) lead by using different OH fields are compared with the uncertainties in $CH_4$ emissions given by Saunois et al. (2016)and Locatelli et al. (2013).

| Study | This study (Impact of OH) | | | | Saunois et al. (2016) | Locatelli et al. (2013) |
|---|---|---|---|---|---|---|
| Period | 2000/07/01-2002/06/01 | | | | 2000-2009 | 2005 |
| Experiment | Inv1 (Original OH) | | Inv2 (Scaled OH) | | TD ensemble | Transport model errors |
| Region | Mean±SD[range[1]] | Unc. | Mean±SD [range] | Unc. | Unc. | Unc. |
| global | 567±34[518-611] | 17% | 551±2[548-555] | 1% | 6% | 5% |
| 60°-90°N | 29±1[27-30] | 12% | 29±1[27-30] | 12% | 50% | 10%(NH) |
| 30°N-60°N | 174±8[158-183] | 14% | 172±6[159-178] | 11% | 20% | 10%(NH) |
| 0°-30°N | 199±14[178-217] | 20% | 192±1[191-194] | 1% | 13% (<30°N) | 10%(NH) |
| 0°-30°S | 147±14[121-167] | 30% | 140±6[133-153] | 14% | 13% (<30°N) | 24%(SH) |
| 30°S-90°S | 19±1[17-20] | 18% | 18±1[18-19] | 9% | 13% (<30°N) | 24%(SH) |
| America | 45±2[42-48] | 11% | 45±1 [42-46] | 8% | 25% | 37% (North America) |
| Canada | 27±1[24-28] | 17% | 27±1 [24-28] | 13% | 70% | 37% (North America) |
| Europe | 27±1 [25-28] | 12% | 27±1 [25-28] | 11% | 43% | 23% |
| Russia | 33±1 [30-35] | 13% | 33±1 [30-34] | 12% | 31% | 38% |
| China | 42±5 [33-50] | 39% | 40±3 [35-43] | 20% | 11% | 25% (Asia) |
| Southeast Asia | 38±3 [34-41] | 20% | 37±0.3 [36-37] | 3% | 42% | 25% (Asia) |
| South Asia | 59±6 [51-66] | 24% | 57±0.8 [56-58] | 4% | 44% | 25% (Asia) |

| | | | | | | |
|---|---|---|---|---|---|---|
| Northern South America | 73±9[58-85] | 37% | 69±4 [65-77] | 17% | 44% | 48% (South America) |
| Southern South America | 33±4[27-39] | 37% | 31±2[29-36] | 20% | 94% | |
| Africa | 76±4 [68-82] | 18% | 74±1 [73-77] | 6% | 42%-45% | 30% |


**Table 4.** Global and latitudinal percentage changes of $CH_4$ reaction weighted [OH] from 2000-2002 to 2007-2009.

| | 90-30°S | 30°S-0° | 0°-30°N | 30°-90°N | Global |
|---|---|---|---|---|---|
| **TransCom** | 0.0% | 0.0% | 0.0% | 0.0% | 0.0% |
| **INCA NMHC** | -0.5% | -0.9% | -0.3% | -0.2% | -0.5% |
| **CESM1-WACCM** | 1.1% | 1.6% | 2.5% | 1.2% | 1.8% |
| **EMAC-L90MA** | -0.1% | 0.1% | 1.3% | 1.1% | 0.7% |
| **GEOSCCM** | -0.3% | 1.1% | 1.4% | 1.0% | 1.0% |
| **MRI-ESM1r1** | -2.0% | 0.2% | 2.4% | 1.7% | 1.1% |

**Table 5.** Global total emission changes (in Tg yr$^{-1}$) from the early 2000s (2000/07/01-2002/06/01) to the late 2000s (2007/07/01-2009/06/01) calculated to identify the effect of OH fields (Inv3－Inv2), of OH fixed to early 2000s (Inv4－Inv2), and the contribution of OH changes from the early to late 2000s to top-down estimated $CH_4$ emissions changes (Inv3－Inv4).

| | Inv3－Inv2 | Inv4－Inv2 | Inv3－Inv4 |
|---|---|---|---|
| TransCom | 17.2 | 17.2 | 0.0 |
| INCA NMHC | 16.6 | 19.3 | -2.7 |
| CESM1-WACCM | 30.0 | 18.5 | 11.5 |
| EMAC-L90MA | 20.4 | 15.3 | 5.1 |
| GEOSCCM | 19.1 | 16.1 | 3.0 |
| MRI-ESM1r1 | 27.8 | 14.3 | 13.5 |
| Mean±SD | 21.9±5.7 | 16.9±1.9 | 5.1±6.4 |


**Table 6.** Global sectoral emission changes (in Tg yr⁻¹) from the early 2000s (2000/07/01-2002/06/01) to the late 2000s (2007/07/01-2009/06/01) (mean±SD and the [min-max] range).

|  | Inv3－Inv2 | Inv4－Inv2 | Inv3－Inv4 | Prior |
|---|---|---|---|---|
| Wetland | -3.5±2.5[-6.4- -0.3] | -5.6±1.3[-6.8- -3.0] | 2.1±3.4[3.3-4.8] | 0.0 |
| Agri-waste | 14.2±2.1[12.2-17.2] | 12.3±0.7[11.1-13.2] | 1.9±2.3[-0.4-5.3] | 19.0 |
| Fossil fuel | 8.7±0.8[8.0-10.1] | 8.1±0.8[7.1-9.6] | 0.6±0.9[-0.1-2.2] | 18.0 |
| Other | 2.4±0.5[2.1-3.1] | 2.0±0.2[1.7-2.2] | 0.5±0.5[-0.1-1.2] | 2.5 |
| Total | 21.9±5.7[16.7-30.0] | 16.8±1.9[14.3-19.3] | 5.1±6.4[-2.5-13.1] | 39.4 |

## Figures

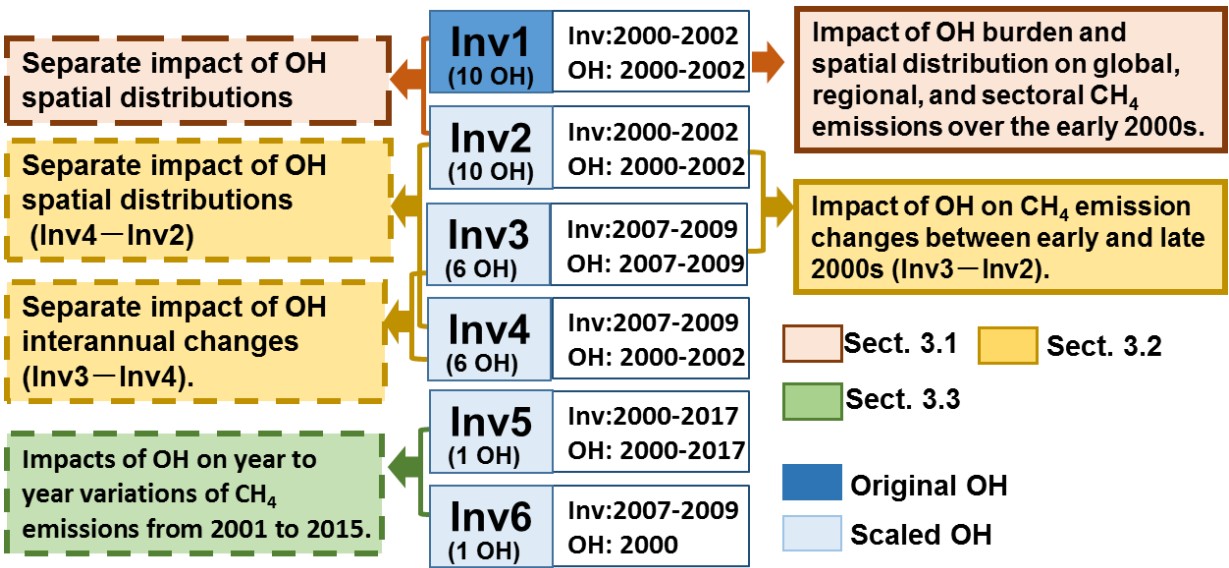

**Figure 1.** A diagram showing inversion experiments (Inv1–Inv6) performed in this study. For each experiment, "Inv" gives the time period of inversion, and "OH" gives the time period of the OH fields used in the inversion. Inv1 is performed using the original OH field, whereas Inv2-Inv5 are performed using scaled OH fields. The colored boxes on the left and right show analyses of inversions we did to examine the OH impacts on inverted CH₄ emissions. The brown, yellow, and green textboxes correspond to analyses presented in Section 3.1, Section 3.2, and Section 3.3, respectively.

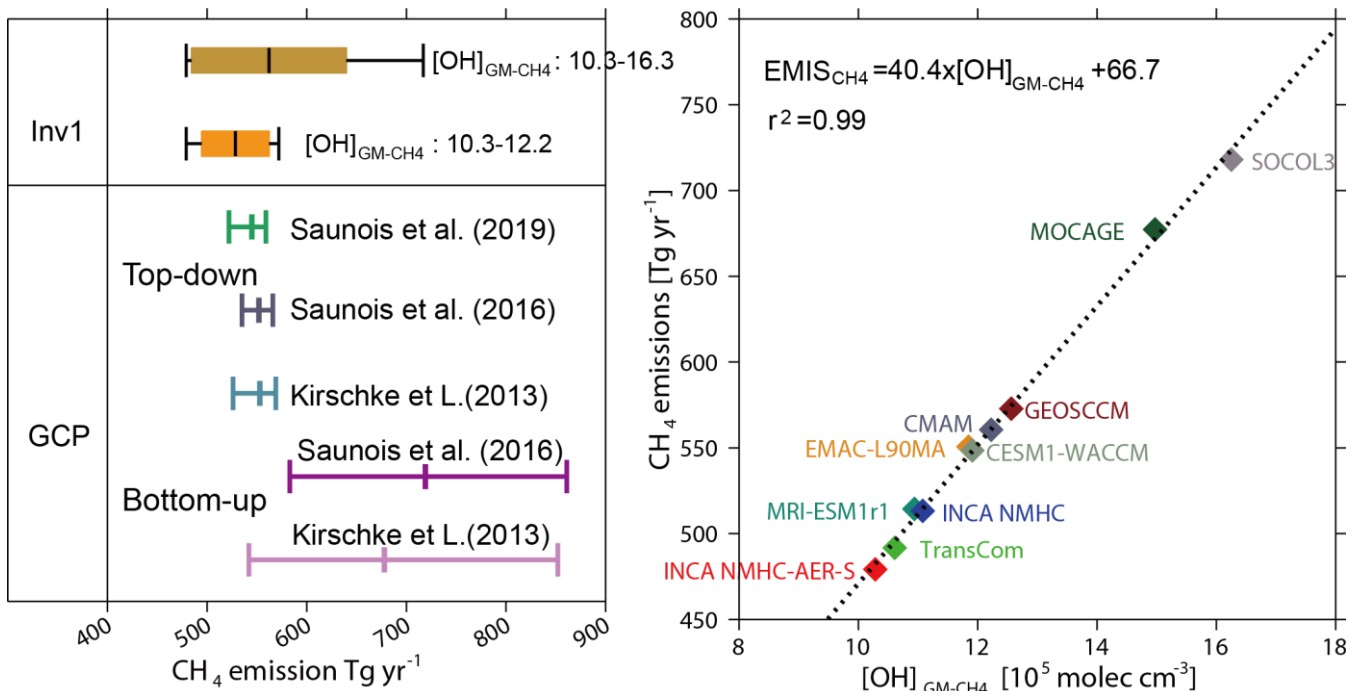

**Figure 2.** Left: The global CH$_4$ emissions from Inv1 (averaged over 2000/07/01-2002/06/01). The bottom-up and top-down estimations over 2000-2009 from previous GCP (Kirschke et al., 2013; Saunois et al., 2016; 2019) are also presented for comparison. The brown box plot shows the inversion results using 10 OH fields, while the orange one shows the results that exclude the largest estimates from the SOCOL3 and MOCAGE OH fields. The left, middle and right whisker chart (vertical lines) represent the minimum, mean, and maximum values of different inversion, and the left and right edges of boxes represent the mean ±one standard deviation. The definition of the box plot is applicable to all those hereafter. Right: The relationship between the optimized CH$_4$ emissions (Tg yr$^{-1}$) in Inv1 and the corresponding [OH]$_{GM-CH4}$ ($\times10^5$ mole cm$^{-3}$). The correlation coefficient (r) and the linear regression equation fitted to the data are shown inset.

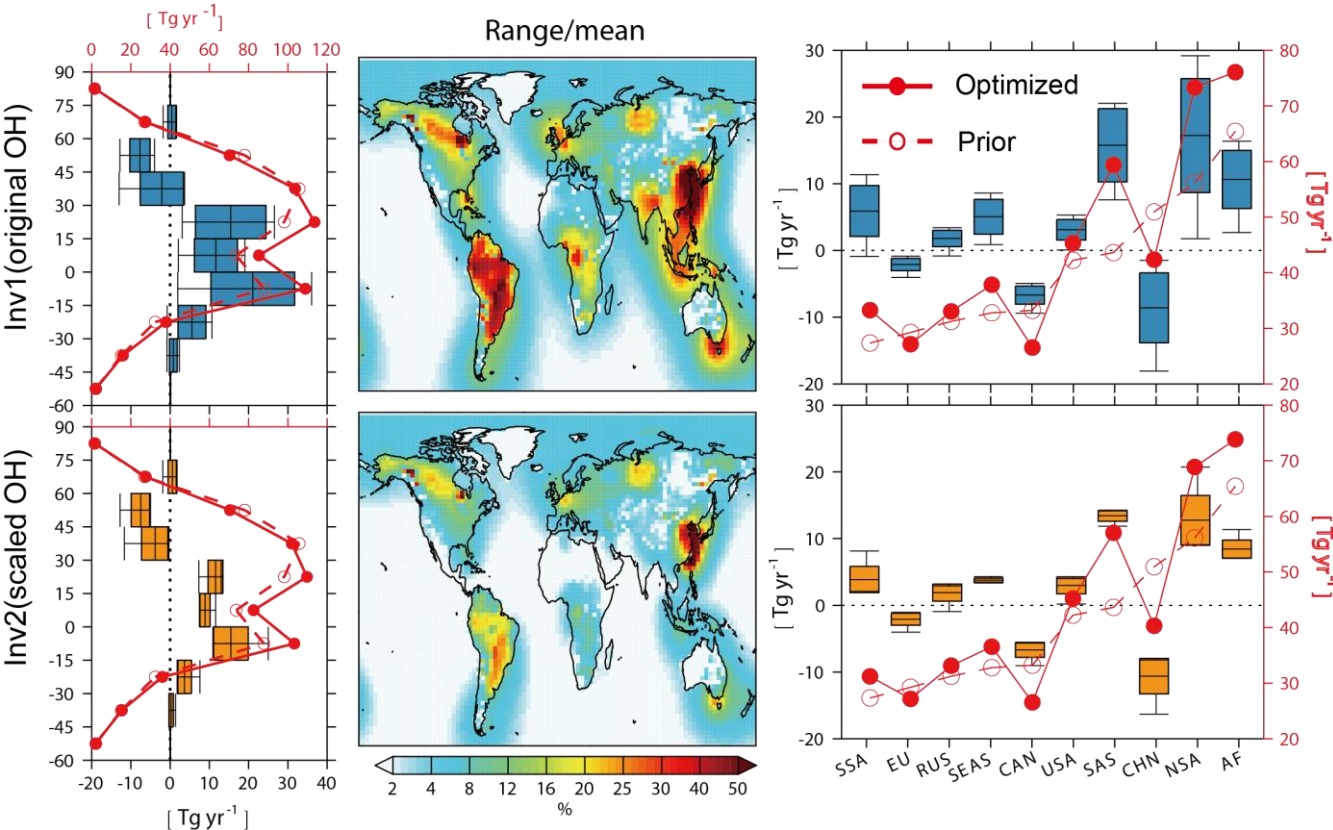

**Figure 3.** Zonal (left), and regional averages (right) of CH$_4$ emissions calculated by Inv1 (top row) and Inv2 (bottom row) with 8 OH fields from 2000/07/01 to 2002/06/01. Left and right panels: prior (dash red line) and mean optimized (solid red line) CH$_4$ emissions for every 15-degree latitudinal band and 10 regions, respectively. Where USA=America, CAN=Canada, EU=Europe, RUS=Russia, CHN=China, SEAS=Southeast Asia, SAS=South Asia, NSA=northern South America, SSA=southern South America, AF=Africa. The full names of the abbreviations are applicable to all figures hereafter. The differences between prior and optimized emissions (optimized minus prior) are shown by the box plots (defined in Fig. 1). Prior and optimized emissions values correspond to the right axes and their differences correspond to the left axes. Middle panel: the ratio of the uncertainty range of emissions estimated with different OH fields (max-min) in each grid-cell calculated by Inv1 (top) and Inv2 (bottom) to multi-inversion mean CH$_4$ emissions.

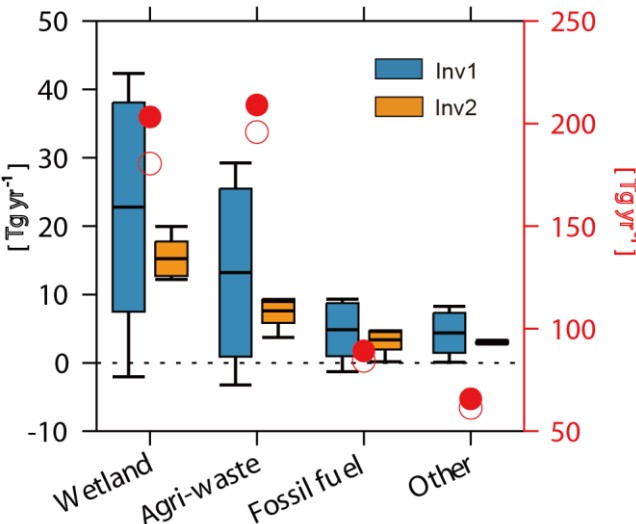

**Figure 4**. Global total $CH_4$ emissions from four broad categories from 2000/07/01-2002/06/01 in Tg yr$^{-1}$. The red circles and dots show the prior emissions and mean optimized emissions, respectively, as calculated by Inv1 (right axes), and the box plots (defined in Fig. 1) show the difference between optimized emissions calculated by Inv1 (blue) and Inv2 (yellow) and prior emissions (optimized minus prior, left axes).

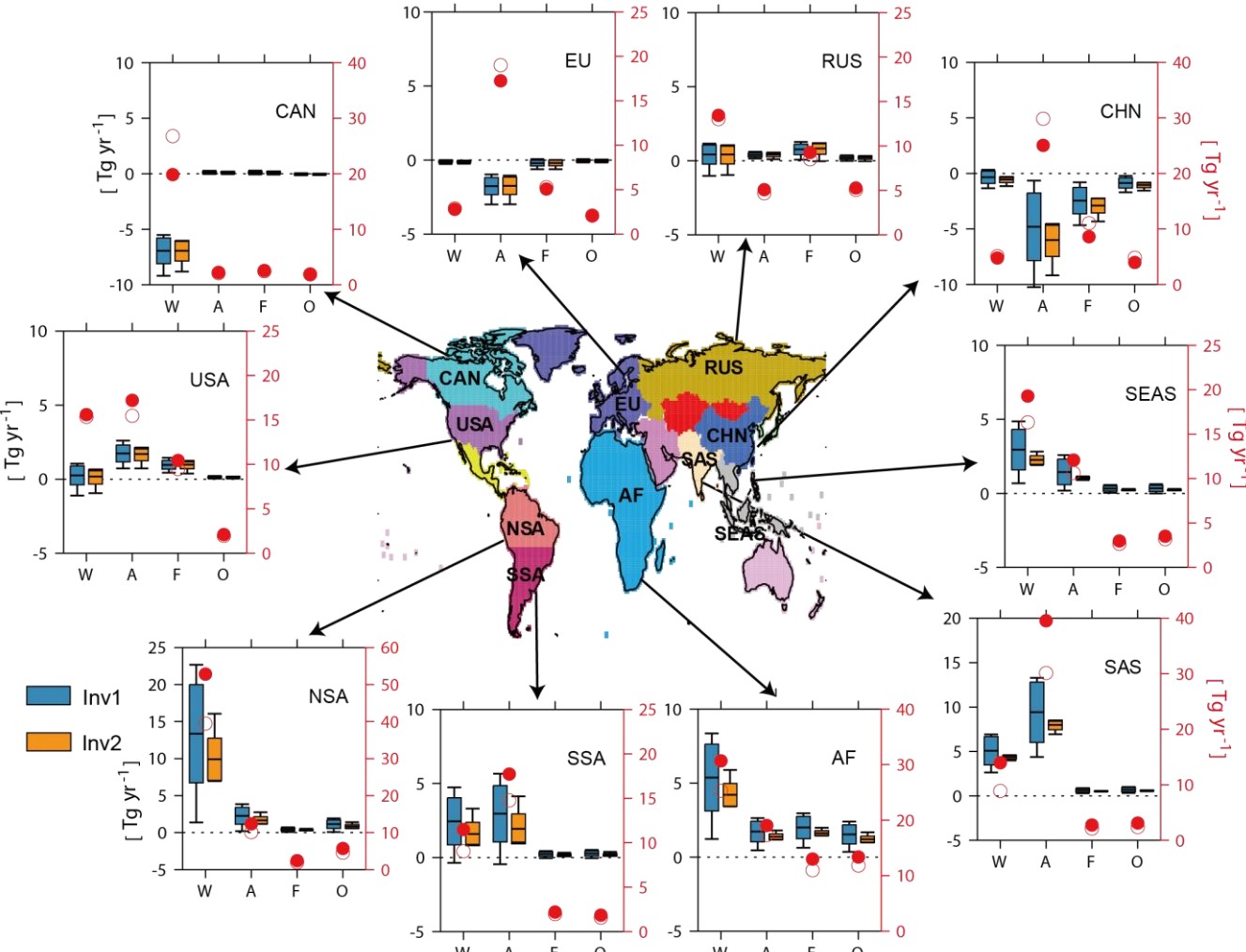

**Figure 5.** Same as Fig. 4 but for prior and optimized emissions over 10 emitting regions covering most of the emitting lands. W=Wetlands, A=Agri-waste, F=Fossil fuels, and O=Others. In Tg yr⁻¹.

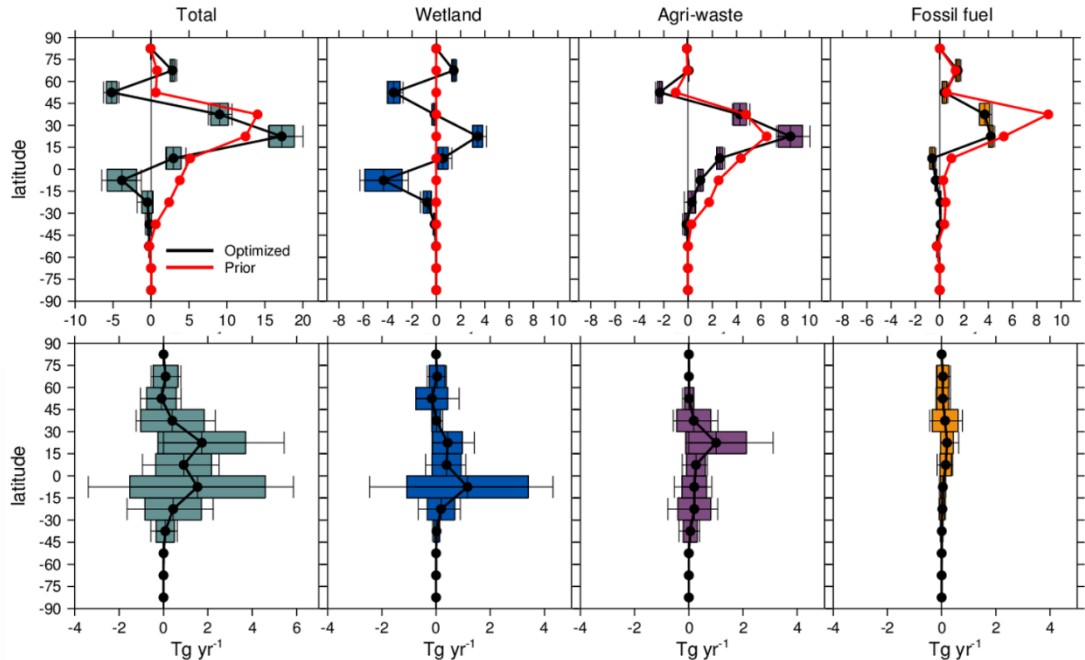

1140

**Figure 6.** Top panel: latitudinal emission (every 15-degree latitudinal band) changes in Tg yr$^{-1}$ from the early 2000s (2000/07/01-2002/06/01) to the late 2000s (2007/07/01-2009/06/01) of total, wetlands, agriculture and waste, and fossil fuel emissions (Inv3－Inv2). Bottom panel: contribution of OH changes on top-down estimated CH$_4$ emission changes between the two periods (Inv3－Inv4). The red lines are changes of prior emissions and the black lines are the mean changes of optimized emissions. The box plots (defined by Fig. 1) show the standard deviations and ranges calculated with different OH fields.

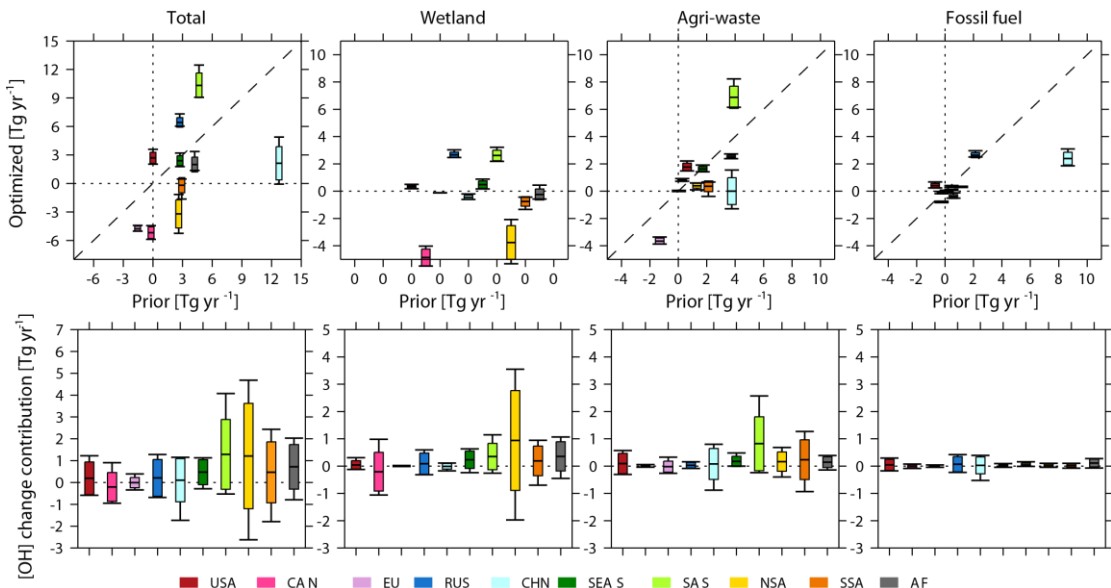

**Figure 7.** Top panel: optimized global total and sectoral regional emission changes in Tg yr$^{-1}$ from the early 2000s to the late 2000s (y-axis) plotted against prior emission changes between the two time periods (x-axis) as derived from Inv2 and Inv3. The prior wetland emissions are constant over time, thus show zero changes (all '0' on the x-axis). Bottom panel: contribution of OH changes between the two periods on top-down estimated emission changes (Inv3－Inv4). The box plots (defined by Fig. 1) show the standard deviations and ranges calculated with different OH fields.

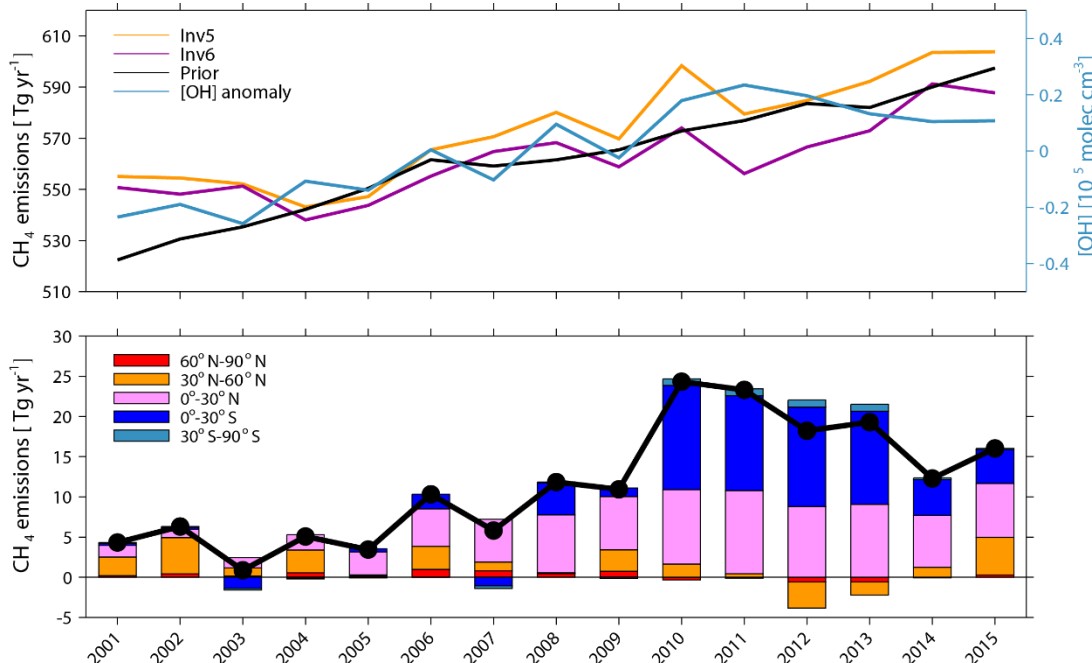

**Figure 8.** Top panel: Time series of global total CH4 emissions calculated by Inv5 (yellow) and Inv6 (purple) plotted together with prior emissions (black), and $[OH]_{GM-CH4}$ anomaly of CESM OH fields (blue). Bottom panel: the difference of global total (black line) and latitudinal (stack bar plots) CH4 emissions between Inv5 and Inv6 (Inv5-Inv6).

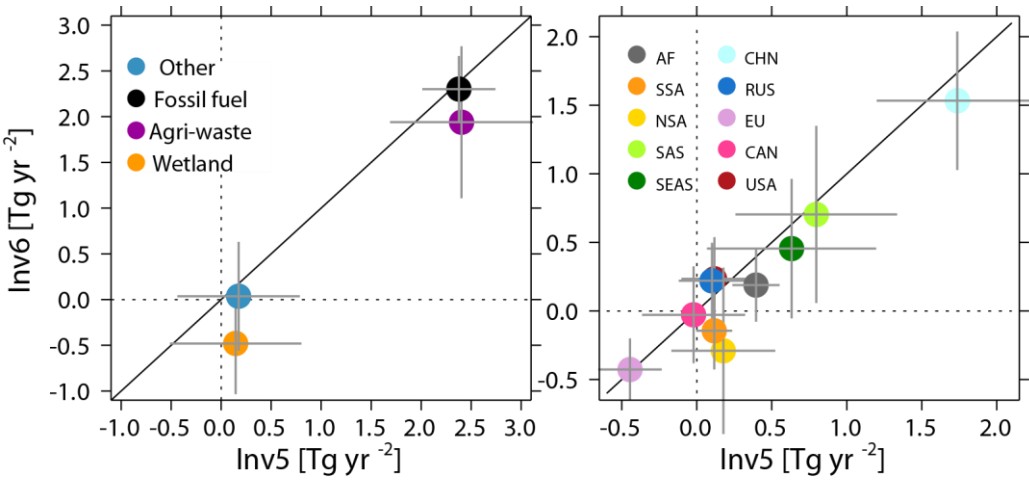

**Figure 9.** Comparison between Inv5 (x-axis) and Inv6 (y-axis) estimated global total CH4 emissions trends in Tg yr$^{-2}$ between 2004 and 2015 for the four categories (left) and over the ten continental regions (right). The error bars show the trend with 95% confidence intervals.