# Peer review of "Influences of hydroxyl radicals (OH) on top-down estimates of the global and regional methane budgets"

_Atmospheric Chemistry and Physics, 2019_

## Short Comment (SC1) · 10 Mar 2020

This study provides important evidence to the importance of improving our estimates of the tropospheric OH sink in other to accurate quantify the $CH_4$ budget. However, I believe that there are three main aspects this study that could make this study have a much stronger impact:

1. The study found the largest absolute OH induced differences for Inv1 over northern South America, South Asia and China and at gridcell level over South America, Central Africa, East and South Asia, and mainly for wetlands, and agriculture and waste. While, it is already explained that the distribution of sampling stations

is one of the reasons for this. This and further reasons for the larger uncertainty in the Tropics were discussed in detail in Bousquet et al., 2011 (another paper from this groups which I think should be reference at that this point in the paper). Furthermore, there are not only less sensitivity to observations in the Tropics, but also larger uncertainty in the fluxes. As a consequence, the inversion fits in everything in the Tropics which is too costly to accommodate elsewhere. Unfortunately, the Tropics is also the region where most of the OH reaction occurs. Therefore, it is very difficult to make conclusions on how the estimation of Tropical fluxes is affected by the OH assumptions on a regional level. I believe the study is missing either one more scenario where the uncertainties for each source are uniform globally, e.g. 5 nmol m$^{-2}$ s$^{-1}$ for wetlands (if there are wetland emissions in the gridcell), and/or include the analysis of the uncertainty reduction and posterior correlations, to determine how well resolved are these regions.

2. The main goal of using an inversion is to find the fluxes that best explain the observations. However, we do not get to see how well the observations are fitted by the inversions with the different OH fields. Therefore, we cannot evaluate which features of the different OH distributions are realistic. By knowing for example the spatial distribution of the residuals, or of the correlations between posterior mixing ratio and observations, we can evaluate if certain spatial patterns are realistic. Also the use of aircraft profiles for validation, e.g. over the Amazon (Miller et al., 2007, Beck et al., 2012, Gatti et al., 2015, Basso et al., 2016), Asia (Brenninkmeijer et al., 2007, Baker et al., 2012, Schuck et al., 2010) or across latitudinal transects (e.g. Wofsy, 2011 and Schuck et al., 2012) could provide information on the realism of the vertical distribution. During the period of the simulation, there were two satellites sensors available SCIAMACHY and IASI with distinctly different sensitivities. SCIAMACHY is more sensitive to the surface, while IASI to the upper troposphere. Using this, it may be possible to say something about how realistic is both the horizontal and vertical distribution

of OH fields. The validation with methylchloroform measurements ($CH_3CCl_3$) as an anthropogenic tracer is also very important, in order to know which magnitude of OH makes sense and which N/S ratio.

3. Link to the validation, there is little discussion on the features of the OH fields provided by the models. For example, Patra et al., (2015) determine that observations of $CH_3CCl_3$ support a N/S gradient of $\sim 1$, so more should be done to explain how probable bias in the modeled OH distributions affects the $CH_4$ estimations. Also many of the different features in the spatial distributions OH are caused by known biases in the climate chemistry models, e.g. the NMVOC levels, the CO burden, CO biases, O3 biases (e.g. Naik et al. 2013, Shindell et al., 2006). Here, it would be very interesting to see, for example, if there is a relationship between the N/S ratio of the OH distributions and the N/S ratio of the posterior fluxes (similar to figure 2). Also, why are SOCOL3 and MOCAGE such outliers?

Some additional comments,

- We are shown inversions with and without interannual variability in the OH fields. However, due to the increase of tropospheric temperatures, even in the simulations with fixed OH or the fields distributed in the TRANSCOM-CH4 experiment, the lifetime of $CH_4$ will decrease. This effect is not quantified in the paper unless I missed it.

- As stated in the study, the transport model uncertainty is very large. This means that the distribution of $CH_4$ is model dependent. Therefore, there could be a large uncertainty in the global OH means weighted by the $CH_4$ reaction. I believe an airmass or volume weighted OH means should be at least provided in the supplement and that the comparison with box models or with other models should be done with air mass or volume weighted means, including the relationship in figure 2.

- On which basis did you choose only 7 of the 20 CCMI simulations?

Minor edits and information display.

- You mentioned the TRANSCOM-2011 project. However, this was actually known as the TRANSCOM-CH4 (Patra et al., 2011), since there have been several TRANSCOM projects mainly with $CO_2$. It might be useful to mention, that in Patra et al., (2011), the OH fields from Spivakovsky (2000) were scaled to match the $CH^3CCl^3$ decay by Marteen Kroll in the TM5 model.

- Table 1 and table 2 are missing the units

- Could you specify which convection parameterization is used? In Locatelli et al., (2015) three parameterizations are used.

- I think that at least in the supplements you should include the maps of the mean differences between the scenarios, e.g. $E_change_all, E_change_fixoh, E_change_varoh$.

- In figures 4 and 5, would it be possible to show the a priori uncertainties as error bars? In general I find the double axes confusing and maybe a single axis with absolute emissions would be better.

References

Baker, A. K., Schuck, T. J., Brenninkmeijer, C. A. M., Rauthe-Schöch, A., Slemr, F., van Velthoven, P. F. J., Lelieveld, J. (2012). Estimating the contribution of monsoon-related biogenic production to methane emissions from South Asia using CARIBIC observations. Geophysical Research Letters, 39(10), n/a–n/a. http://doi.org/10.1029/2012GL051756

[Figure]

Basso, L. S., Gatti, L. V., Gloor, M., Miller, J. B., Domingues, L. G., Correia, C. S. C., Borges, V. F. (2016). Seasonality and interannual variability of CH4 fluxes from the eastern Amazon Basin inferred from atmospheric mole fraction profiles. Journal of Geophysical Research-Atmospheres, 121(1), 168–184. http://doi.org/10.1002/2015JD023874

Beck, V., Chen, H., Gerbig, C., Bergamaschi, P., Bruhwiler, L. M. P., Houweling, S., et al. (2012). Methane airborne measurements and comparison to global models during BARCA. Journal of Geophysical Research-Atmospheres, 117(D). http://doi.org/10.1029/2011JD017345

Brenninkmeijer, C. A. M., Crutzen, P. J., Boumard, F., Dauer, T., Dix, B., Ebinghaus, R., et al. (2007). Civil Aircraft for the regular investigation of the atmosphere based on an instrumented container: The new CARIBIC system. Atmospheric Chemistry and Physics, 7(18), 4953–4976.

Bousquet, P., Ringeval, B., Pison, I., Dlugokencky, E. J., Brunke, E. G., Carouge, C., et al. (2011). Source attribution of the changes in atmospheric methane for 2006-2008. Atmospheric Chemistry and Physics, 11(8), 3689–3700. http://doi.org/10.5194/acp-11-3689-2011

Gatti, L. V., Gloor, M., Miller, J. B., Doughty, C. E., Malhi, Y., Domingues, L. G., et al. (2015). Drought sensitivity of Amazonian carbon balance revealed by atmospheric measurements. Nature, 506(7486), 76–80. http://doi.org/10.1038/nature12957

Locatelli, R., Bousquet, P., Saunois, M., Chevallier, F., Cressot, C. (2015). Sensitivity of the recent methane budget to LMDz sub-grid-scale physical parameterizations. Atmospheric Chemistry and Physics, 15(17), 9765–9780. http://doi.org/10.5194/acp-15-9765-2015

Miller, J. B., Gatti, L. V., d'Amelio, M. T. S., Crotwell, A. M., Dlugokencky, E. J., Bakwin, P. S., et al. (2007). Airborne measurements indicate large methane emissions

from the eastern Amazon basin. Geophysical Research Letters, 34(10), L10809–. http://doi.org/10.1029/2006GL029213

Naik, V., Voulgarakis, A., Fiore, A. M., Horowitz, L. W., Lamarque, J. F., Lin, M., et al. (2013). Preindustrial to present-day changes in tropospheric hydroxyl radical and methane lifetime from the Atmospheric Chemistry and Climate Model Intercomparison Project (ACCMIP). Atmospheric Chemistry and Physics, 13(1), 5277–5298. http://doi.org/10.5194/acp-13-5277-2013

Patra, P. K., Houweling, S., Krol, M., Bousquet, P., Belikov, D., Bergmann, D., et al. (2011). TransCom model simulations of CH4 and related species: linking transport, surface flux and chemical loss with CH4 variability in the troposphere and lower stratosphere. Atmospheric Chemistry and Physics, 11(24), 12813–12837. http://doi.org/10.5194/acp-11-12813-2011

Patra, P. K., Krol, M. C., Montzka, S. A., Arnold, T., Atlas, E. L., Lintner, B. R., et al. (2015). Observational evidence for interhemispheric hydroxyl-radical parity. Nature, 513(7517), 219–223. http://doi.org/10.1038/nature13721

Shindell, D. T., Faluvegi, G., Stevenson, D. S., Krol, M. C., Emmons, L. K., Lamarque, J. F., et al. (2006). Multimodel simulations of carbon monoxide: Comparison with observations and projected near-future changes. Journal of Geophysical Research-Atmospheres, 111(D), D19306. http://doi.org/10.1029/2006JD007100

Schuck, T. J., Brenninkmeijer, C. A. M., Baker, A. K., Slemr, F., Velthoven, von, P. F. J., Zahn, A. (2010). Greenhouse gas relationships in the Indian summer monsoon plume measured by the CARIBIC passenger aircraft. Atmospheric Chemistry and Physics, 10(8), 3965–3984.

Schuck, T. J., Ishijima, K., Patra, P. K., Baker, A. K., Machida, T., Matsueda, H., et al. (2012). Distribution of methane in the tropical upper troposphere measured by CARIBIC and CONTRAIL aircraft. Journal of Geophysical Research, 117(D19), n/a–

n/a. http://doi.org/10.1029/2012JD018199

Wofsy, S. C. (2011). HIAPER Pole-to-Pole Observations (HIPPO): fine-grained, global-scale measurements of climatically important atmospheric gases and aerosols. Philosophical Transactions of the Royal Society a: Mathematical, 369, 2073–2086. http://doi.org/10.1098/rsta.2010.0313

---

## Referee Comment (RC1) · Anonymous Referee #1 · 16 Mar 2020

Zhao et al. assess systematically how uncertainties of OH concentrations affects our inference of the global and regional methane emissions and their decadal changes from the existing surface measurement network. The authors performed a series of inversion experiments using varied OH fields and used the standard deviations of an inversion ensemble to represent the uncertainty due to OH fields. The work is very important, as the uncertainty source of prescribed OH fields have not been quantitatively assessed in previous syntheses (e.g., Saunois et al. 2017, 2019). However, the manuscript can be improved with better presentation and in-depth discussion. I'd recommend the publication of this manuscript if the following issues are addressed.

[Figure]

1. The manuscript lacks quantitative comparisons of the results with other uncertainty sources (as assessed in literature) of methane emission estimations. The comparison could provide readers both the context and the insight. For example, I am looking for answers to the following questions: how large is the uncertainty due to OH compared to other uncertainty sources (e.g., transport)? Is the uncertainty due to OH the bottle-neck for understanding the global and regional methane budget? In which regions, the uncertainty due to OH dominates; and in which regions, they are not that important? Is it adequate to reduce the uncertainty of global mean OH for the purpose of improving estimates for global and regional methane emissions? Or reducing uncertainty in OH spatial distribution is equally important? These questions are interesting to readers and can be addressed by putting the results of this paper in the context of literature (such as Saunois et al. 2017 from the authors' group).

2. The regional results are specific to the observing system (i.e., NOAA surface network). Surface observations are relatively dense in North America and West Europe, but very sparse near South America, Tropical Africa, and Tropical Asia. Therefore, the inversion tends to adjust emissions from regions less constrained by observations, if any global mismatch exists, leading to large spread of estimates in these regions. Inclusion of more observations may lead to different spatial patterns in Fig. 3. It is important to acknowledge that the conclusion about regional emissions applies to only this specific observing system. The authors mentioned site locations when explaining the difference between Inv1 and Inv2; however results from other experiments may also be explained by this factor, at least partly. In addition, OH concentrations are highest over tropics, therefore, it is expected that the difference in OH from varied fields is largest over tropics. This could explain the larger posterior flux range in tropics for Inv1 (Fig. 3).

Other comments

38-40: The sentence reads awkward. Physically, increases in OH burden cannot contribute to increases in emissions. Clarify or rephrase to avoid any confusion.

53-54: The word "additional" is confusing here.

71-72: Unclear what "catalytic chemistry" in this sentence is referred to. Also, the statement "a small perturbation of OH can result in significant change in atmospheric CH4" is inaccurate or ambiguous. The author may want to say "... significant change in the budget (or budget imbalance) of atmospheric CH4".

72-75: There are other OH sources such as O3+HO2, H2O2 photolysis, and OVOCs photolysis that become important depending on the chemical environment, for example, see Lelieveld et al. (2016).

Lelieveld, J., Gromov, S., Pozzer, A., and Taraborrelli, D.: Global tropospheric hydroxyl distribution, budget and reactivity, Atmos. Chem. Phys., 16, 12477–12493, https://doi.org/10.5194/acp-16-12477-2016, 2016.

78: ÂăDirect measurement of OH is challenging but possible. But estimates of global mean from sparse direct measurements is nearly impossible because the large variation of OH as a result of its short lifetime.

101-103: Optimizations of CH4 emissions together with OH concentrations have been done using 3-D model inversions (e.g., Cressot et al., 2014, Zhang et al., 2018 and Maasakkers et al., 2019), in addition to two-box model analysis. These studies all used satellite data though.

Cressot, C. et al. On the consistency between global and regional methane emissions inferred from SCIAMACHY, TANSO-FTS, IASI and surface measurements, Atmos. Chem. Phys., 14, 577–592, https://doi.org/10.5194/acp-14-577-2014, 2014.

Zhang, Y. et al. Monitoring global tropospheric OH concentrations using satellite observations of atmospheric methane, Atmos. Chem. Phys., 18, 15959–15973, https://doi.org/10.5194/acp-18-15959-2018, 2018.

Maasakkers et al. Global distribution of methane emissions, emission trends, and OH concentrations and trends inferred from an inversion of GOSAT satellite data for

2010–2015, Atmos. Chem. Phys., 19, 7859–7881, https://doi.org/10.5194/acp-19-7859-2019, 2019

Section 2.1: from the description, it is not clear if these OH fields vary from month to month, or are annual mean 3-D fields.

Line 155: What temperature field do you use to compute [OH]GM-CH4 for different models? And how "troposphere" is defined in this calculation? Line 158: Is latitudinal distribution of OH also a factor (and maybe even more important factor) that results in [OH]GM-CH4 > [OH]GM-M ?

Eq. 1. The (x-xb) term is repeated twice.

Line 172: Since only emissions are optimized in the inversion, it's a bit misleading to say H(x) represents sensitivity to sinks.

Line 205: What about Cl?

Line 231-232: "To separate the influence of OH spatial distributions from that of global mean [OH]. . .". As commented above, it is unclear whether the OH fields vary monthly or annual mean. If the former, then in addition to influence of spatial distribution, the influence of seasonal variation is also embedded. If the latter, then the study design has a major flaw because the latitudinal distribution of OH has a pronounced seasonal cycle.

Line 239-240: Please denote inv3 and inv4 explicitly after 2007-2009 and 2000-2002 to make it easier to follow.

Line 241-243: I don't think Inv4-Inv2 represents the impact of OH spatial distribution.

Line 250: Which one has the largest trend, which may be more relevant in this setting?

Line 269-273: This sentence does not flow smoothly within the context (results from Inv1). Remove it or move it somewhere else.

Line 278: Not clear to me how this helps decreasing discrepancies with bottom-up estimates? Fig. 2 does not show the discrepancies are reduced to me. Please clarify. Also please provide the values (and ranges) of bottom-up estimates in the text for a clear comparison.

Line 284-290: Is it possible that the difference is due to the fact that [OH]GM-CH4 is used here instead of [OH]GM-M (which I assume was used in these studies)? It does not convince me that the difference is due to the inter-hemispheric transport and stratospheric loss in 3-D model vs. 2-box model. Choices of hemispheric mean reaction rate of OH+CH4 can also introduce biases in 2-box model.

Line 316: Does the seasonality of OH fields also play a role here?

Line 338: Please explicitly state which uncertainty sources Saunois et al. (2016) considered. The comparison may be misleading otherwise.

Line 343-345: Likely because these regions have high prior emissions, but are not well constrained by surface measurements. So, it should be stated that these regional features are not intrinsic of the atmosphere, but specific to the observing system of interest.

Line 355: what is the "total differences"? How large are they?

Line 364-367: I don't understand the logic here. I think it is probably related to OH concentration being much higher in tropics than extra-tropics.Ăă

Line 379: The range of global total CH4 emissions by Inv2 (551+-2 Tg a-1) should be reported and discussed in 3.3.1, in comparison with Inv1.

Line 418: Be clearer what "global scale increase" in this sentence is referred to. It is ambiguous in the current form.

Line 485-489: The assessment of the uncertainty due to OH fields relative to other uncertainty sources are too qualitative throughout the manuscript. More insight can

[Figure]

be gained by quantitatively comparing to uncertainty estimates in literature such as Saunois et al.

Table 2 Quite confusing. Why global and hemispheric emissions are only shown for Inv1, but the inter-hemispheric differences are shown for both Inv1 and Inv2? Also, unit should be denoted in the caption.

Table 5 With fixed OH field, you still expect an increasing OH sink (and therefore increasing emissions) because of increasing CH4 concentration and temperature. This should be clarified somewhere in the text.

Fig. 2 The R2=0.99 line in the right panel: it should be acknowledged that other sinks of methane (such as soil absorption, Cl, and stratospheric loss) are not optimized and are specified with the same field in these inversions. Uncertainty in these sinks, if considered, will certainly create some spread in the data.

Fig. 3 To interpret this figure, the author should consider the uneven sampling of the surface network. The ranges of inferred regional emissions are large where observations are sparse, because it "costs" the least for the inversion to adjust in these regions. The inference for regional emissions is specific to the particular observing system. Having more surface stations in the southern hemisphere, or including satellite observations, would change the spatial pattern shown in this figure.

---

## Referee Comment (RC2) · Anonymous Referee #2 · 8 Apr 2020

**1 Overview:**

Review of "*Influences of hydroxyl radicals (OH) on top-down estimates of the global and regional methane budgets*" by Zhao *et al.*

This review slipped through the cracks as the COVID-19 situation evolved here. My sincere apologies for any hold ups.

Zhao *et al.* present an analysis of a set of methane inversions using a set of 10 different OH fields from the CCMI experiment. They find the magnitude of methane emissions differs by roughly 30% (518-757 Tg/yr) depending on what OH fields they use. Over-

all, the study is useful in quantifying some of the uncertainties in methane emission estimates due to uncertain OH concentrations. The main shortcoming is the lack of discussion of what actually causes some of the differences (or really any discussion of OH). The figures are high quality but the text is very hard to follow because it's filled with many acronyms and parenthetical expressions. I would recommend major revisions.

**2   Comments:**

**2.1   What processes actually drive these differences?**

The main issue I feel is totally missing from the manuscript is any discussion of what processes are actually driving some of these differences. I believe there was only a single paragraph (Lines 70-79) even mentioning anything about what might affect OH. For example, do some of the CCMI models or inversions show consistent patterns with known climate oscillations? This was surprising given that this is a paper focused on how OH impacts methane. The obvious question is what leads to these differences/similarities in OH. The authors seem to treat the CCMI models as a black box which makes it hard to gain any understanding of what's happening. Given this, the only major take-away I had from the paper is that *"OH can lead to big differences in methane estimates"*, but this was already demonstrated by the box modeling papers (and others) that Zhao et al. are highly critical of. For example, the Rigby paper had error bars on their OH fields that bounded zero and the Turner paper had a case where OH didn't change. Both of these led to radically different methane emissions. Back to my point, I would find this manuscript much more useful and compelling if the authors actually highlighted processes and phenomena that lead to similar methane inversion responses. From Holmes et al., ACP (2013; https://doi.org/10.5194/acp-13-285-2013) we know some of the major processes that influence OH and Turner et al., PNAS (2018; https://doi.org/10.1073/pnas.1807532115) showed how this can co-vary

with things like ENSO, do the authors see ENSO signals in the methane inversions? A recent paper from Nguyen et al., GRL (2020) tried to look at these feedbacks in a simple model. The authors should at least touch on the processes that influence OH, particularly those that could also influence methane.

**2.2  Oversight of previous work and faith in the CCMI models**

The authors seem to have quite a bit of faith in the CCMI models, more than this reviewer finds to be justified. There are quite a few known shortcomings of the models. For example, the models don't even get the ratio of the N/S gradient in OH correct. Yet the authors are quick to criticize MCF-constrained [OH] fields with seemingly no validation of their own OH fields (e.g., Lines 595-600). Is their analysis consistent with MCF? The authors seem to be arguing that these model-derived forward simulations of OH are more reliable than reconstructions.

The strongest claims made in this paper seem to be those that are critical of previous work estimating OH (e.g., Rigby and Turner). For example, Lines 595-600, the abstract is dismissive of box modeling results: 'previous research mostly relied on box modeling inversions with a very simplified atmospheric transport". The latter line in the abstract isn't even correct as there has been quite a bit of non-box model work the authors seem to discount or miss: McNorton et al., ACP (2016; https://doi.org/10.5194/acp-16-7943-2016), Gaubert et al., GRL (2017; https://doi.org/10.1002/2017GL074987), Rigby et al., PNAS (2017; https://doi.org/10.1073/pnas.1616426114), Turner et al., PNAS (2017; https://doi.org/10.1073/pnas.1616020114), McNorton et al., ACP (2018; https://doi.org/10.5194/acp-18-18149-2018), Maasakkers et al., ACP (2019; https://doi.org/10.5194/acp-19-7859-2019), Naus et al., ACP (2018; https://doi.org/10.5194/acp-19-407-2019), Nguyen et al., (2020; https://doi.org/10.1029/2019GL085706), and He et al., ACP (2020; https://doi.org/10.5194/acp-20-805-2020).  About half of these papers use 3-D

atmospheric transport models and some even include fully-coupled chemistry (e.g., He et al., 2020), which is more comprehensive than the models used by the authors. The authors should do a more complete reading of the literature as they don't cite Holmes et al., ACP (2013), Murray et al., ACP (2014), or any of Michael Prather's papers.

2.3   Very difficult to follow

The paper is filled with jargon and abbreviations. For example, nearly half of the text in Lines 440-452 are acronyms or parenthetical expressions interjecting things. This was very hard to follow as a reader.

---

## Author Comment (AC1) · 12 May 2020

**Reply to SC1: ' 'Important evidence to importance of OH but it can have more impact' '**

**Comments**: This study provides important evidence to the importance of improving our estimates of the tropospheric OH sink in other to accurate quantify the CH4 budget. However, I believe that there are three main aspects this study that could make this study have a much stronger impact:

**Response: We thank Tonatiuh Guillermo Nuñez Ramirez for the helpful comments. Please see out itemized responses below.**

**Comments:** 1. The study found the largest absolute OH induced differences for Inv1 over northern South America, South Asia and China and at gridcell level over South America, Central Africa, East and South Asia, and mainly for wetlands, and agriculture and waste. While, it is already explained that the distribution of sampling stations is one of the reasons for this. This and further reasons for the larger uncertainty in the Tropics were discussed in detail in Bousquet et al., 2011 (another paper from this groups which I think should be reference at that this point in the paper). Furthermore, there are not only less sensitivity to observations in the Tropics, but also larger uncertainty in the fluxes. As a consequence, the inversion fits in everything in the Tropics which is too costly to accommodate elsewhere. Unfortunately, the Tropics is also the region where most of the OH reaction occurs. Therefore, it is very difficult to make conclusions on how the estimation of Tropical fluxes is affected by the OH assumptions on a regional level. I believe the study is missing either one more scenario where the uncertainties for each source are uniform globally, e.g. 5 nmol m-2 s-1 for wetlands (if there are wetland emissions in the gridcell), and/or include the analysis of the uncertainty reduction and posterior correlations, to determine how well resolved are these regions.

**Response:** We discussed the impact of the distribution of the sampling stations in the text(L407-415):**

"The uncertainties in the top-down estimated regional emissions are not only due to inter-model

differences of the regional OH fields but also rely on the distribution of the surface observations used in the inversions. Over the regions with large prior emissions but less constrained by observations (e.g. South America, South Asia, and China), our OH analysis leads to larger uncertainties than regions that are well constrained by observations (e.g. the North America and Canada) (Fig. S3). The results may indicate that on the regional scale, the top-down estimated CH4 emissions and the uncertainties lead by OH are specific to the observation system retained. If more surface observations (e.g. in the southern hemisphere) or satellite columns with a more even global coverage were included in our inversions, spatial patterns of the top-down estimated CH4 emissions and their uncertainties (as shown by Fig.3) could be different."

We acknowledge the fact that more scenarios could provide additional conclusions but this would necessitate extensive additional work and the paper is already long. Here this study aims to quantify the uncertainties in the current top-down due to uncertainties in OH. Analysis of how the top-down inversion can resolve the regional emissions by testing the uncertainty reduction and posterior correlations can be a separate study in our further study. However, we thank you for the suggestion and keep the idea for future works.

**Comments: 2.** The main goal of using an inversion is to find the fluxes that best explain the observations. However, we do not get to see how well the observations are fitted by the inversions with the different OH fields. Therefore, we cannot evaluate which features of the different OH distributions are realistic. By knowing for example the spatial distribution of the residuals, or of the correlations between posterior mixing ratio and observations, we can evaluate if certain spatial patterns are realistic. Also the use of aircraft profiles for validation, e.g. over the Amazon (Miller et al., 2007, Beck et al., 2012, Gatti et al., 2015, Basso et al., 2016), Asia (Brenninkmeijer et al., 2007, Baker et al., 2012, Schuck et al., 2010) or across latitudinal transects (e.g. Wofsy, 2011 and Schuck et al., 2012) could provide information on the

realism of the vertical distribution. During the period of the simulation, there were two satellites sensors available SCIAMACHY and IASI with distinctly different sensitivities. SCIAMACHY is more sensitive to the surface, while IASI to the upper troposphere. Using this, it may be possible to say something about how realistic is both the horizontal and vertical distribution.

**Response:**

In the updated version, we evaluate the inversions using aircraft observations. Usually, one can use the surface observations to evaluate the inversions using satellite data but we do not use satellite data to evaluate the inversions using surface observations. Comparison can still be made but (i) the observations from IASI do not provide the averaging kernel, thus they cannot directly compare with model simulations, and (ii) SCIAMACHY CH4 data experience significant to large systematic errors, limiting strongly the interest of comparison.

**We added in the text (L284-L296):**

"We evaluate the optimized CH4 emissions by comparing the simulated CH4 mixing ratios using prior and posterior CH4 emissions with independent measurements from the NOAA/ESRL Aircraft Project. The location of the observation site (Table S1) and the vertical profile of the model bias in CH4 mixing ratios compared with the aircraft observations (model minus observations) are shown in the supplement (Fig. S4a for Inv1 and Fig. S4b for Inv2). The comparisons with independent aircraft observations confirm the improvement of model-simulated CH4 mixing ratios when using posterior emissions. All of the inversions in Inv1 and Inv2 reach small biases when compared with aircraft observations (right panel of Fig.S4a and Fig.S4b), which means that it is hard to distinguish which OH spatial and vertical distributions are more realistic in terms of quality of fit to these aircraft CH4 observations. For Inv1, the root mean square errors (RMSE =  $\frac{\sqrt{\Sigma(model-observation)^2}}{n_obs}$ , n obs is the number of observations) are reduced from up to more than 100ppby (prior) emissions to ~10ppbv (posterior). For Inv2, although the CH4 mixing ratios simulated using prior emissions already match well with aircraft observations (MSE=8-17ppbv), the posterior emissions still reduce the RMSE by up to 10ppbv. "

We added Table S1 and Figure S4a and Figure 4b in the supplement:

| STATION | SITE LOCATION      | воттом | ТОР    | latitude  | longitude  |
|---------|--------------------|--------|--------|-----------|------------|
| ID      |                    | ALT(m) | ALT(m) |           |            |
| CAR     | Briggsdale, Co     | 1658   | 11879  | 40° 22' N | 104° 17' W |
| HAA     | Molokai Island, HI | 305    | 8104   | 21° 14' N | 158° 57' W |
| HFM     | Harvard Forest, Ma | 582    | 8063   | 42° 32' N | 72° 10' W  |
| PFA     | Poker Flat, AK     | 131    | 7604   | 65° 04' N | 147° 17' W |

Table S1. Location of the NOAA ESRL aircraft sites.

---

## Author Comment (AC2) · 12 May 2020

*Reply to RC1: ' Review of "Influences of hydroxyl radicals (OH) on top-down estimates of the global and regional methane budgets" '*

**Comment:** Zhao et al. assess systematically how uncertainties of OH concentrations affects our inference of the global and regional methane emissions and their decadal changes from the existing surface measurement network. The authors performed a series of inversion experiments using varied OH fields and used the standard deviations of an inversion ensemble to represent the uncertainty due to OH fields. The work is very important, as the uncertainty source of prescribed OH fields have not been quantitatively assessed in previous syntheses (e.g., Saunois et al. 2017, 2019). However, the manuscript can be improved with better presentation and in-depth discussion. I'd recommend the publication of this manuscript if the following issues are addressed.

**Response: We thank the reviewer for his/her helpful comments. All of them have been addressed in the revised manuscript. Please see out itemized responses below.**

**Comments:** 1. The manuscript lacks quantitative comparisons of the results with other uncertainty sources (as assessed in literature) of methane emission estimations. The comparison could provide readers both the context and the insight. For example, I am looking for answers to the following questions:

1 how large is the uncertainty due to OH compared to other uncertainty sources (e.g., transport)?

**Response:**

**For the uncertainties in global total $CH_4$ emissions lead by OH, we added the values in section 3.1.1 (L316-326):**

[revised manuscript text omitted]

For emissions changes during the 2000s, we added in section 3.2.2 (L555-566):

" We now compare the uncertainty of top-down estimated $CH_4$ emission changes from the early to the late 2000s due to different OH spatial-temporal variations with that ensemble of top-down studies given by Saunois et al. (2017). For the sectoral emissions, the emission changes from agriculture and waste and from wetland show the largest uncertainties (more than 50% of multi-inversions mean, Inv3−Inv2 in Table 6) induced by OH spatial-temporal variations, comparable to that given by Saunois et al. (2017). On the contrary, the uncertainty of fossil fuel emission changes (24% of multi-inversions mean) is much smaller than that given by Saunois et al. (2017). For regional $CH_4$ emission changes, the uncertainty induced by OH spatial-temporal variations is usually larger than the multi-inversion mean emission changes (except South Asia) and similar to that given by Saunois et al. (2017). The large differences existing in different top-down estimated regional and sectoral emission changes are mainly attributed to model transport errors in Saunois et al. (2017). Here, our results show that uncertainties due to OH spatio-temporal variations can lead to similar biases in top-down estimated $CH_4$ emission changes."

2 Is the uncertainty due to OH the bottleneck for understanding the global and regional methane budget? In which regions, the uncertainty due to OH dominates; and in which regions, they are not that important?

**Response:**

**For the global CH$_4$ budget, in the conclusions and discussion, we have demonstrated (L640-L645):**

**" Based on the ensemble of 10 original OH fields ([OH]$_{GM-CH4}$:10.3-16.3$\times10^5$ molec cm$^{-3}$), the global total CH$_4$ emissions inverted by our system vary from 518 to 757Tg yr$^{-1}$ during the early 2000s, similar to the CH$_4$ emission range estimated by previous bottom-up syntheses and larger than the range reported by the top-down studies (Kirschke et al., 2013; Saunois et al, 2016). The top-down estimated global total CH$_4$ emission varies linearly with [OH]$_{GM-CH4}$, which indicates that at the global scale, a small uncertainty of 1$\times10^5$ molec cm$^{-3}$ (10%) [OH]$_{GM-CH4}$ can result in 40.4Tg yr$^{-1}$ uncertainties in optimized CH$_4$ emissions."**

**For the regional CH$_4$ budget, we added in "Conclusions and discussion" (L647-L654) :**

**"At regional scale (excluding the two highest OH fields), CH$_4$ emission uncertainties due to different OH global burdens and distributions are largest over South America (37% of multi-inversion mean), South Asia (24%), and China (39%), resulting in significant uncertainties in optimized emissions from the wetland and agriculture and waste sectors. These uncertainties are comparable in these regions with those due to model transport errors and inversion system set-up (Locatelli et al., 2013; Saunois et al., 2016). For these regions, the uncertainty due to OH is critical for understanding their methane budget. In other regions, OH leads to smaller uncertainties compared to that given by Locatelli et al. (2013) and Saunois et al. (2016). "**

Is it adequate to reduce the uncertainty of global mean OH for the purpose of improving estimates for global and regional methane emissions? Or reducing uncertainty in OH spatial distribution is equally important?

**Response: We added in Section 4 "Conclusions and discussion" (L705-L713):**

**" Our results indicate that OH spatial distributions, which are difficult to obtain from proxy observations (e.g. MCF), are equally important as the global OH burden for constraining CH$_4$ emissions over mid- and high-latitude regions. Constraining global annual mean OH based on proxy observations (e.g. Zhang et al., 2018; Maasakkers et al., 2019) provides a constraint on global total methane emissions, through the necessity of balancing the global budget (sum of source – sum of sinks = atmospheric growth rate). It also largely reduces uncertainties in optimized CH$_4$ emissions due to OH over most of the tropical regions but not over South America and overall mid-**

**high latitude regions. Also, the spatial and seasonal distributions of OH is found critical to properly infer temporal changes of regional and sectoral CH₄ emissions."**

These questions are interesting to readers and can be addressed by putting the results of this paper in the context of literature (such as Saunois et al. 2017 from the authors' group).

**Comments:** 2. The regional results are specific to the observing system (i.e., NOAA surface network). Surface observations are relatively dense in North America and West Europe, but very sparse near South America, Tropical Africa, and Tropical Asia. Therefore, the inversion tends to adjust emissions from regions less constrained by observations, if any global mismatch exists, leading to large spread of estimates in these regions. Inclusion of more observations may lead to different spatial patterns in Fig. 3. It is important to acknowledge that the conclusion about regional emissions applies to only this specific observing system. The authors mentioned site locations when explaining the difference between Inv1 and Inv2; however results from other experiments may also be explained by this factor, at least partly. In addition, OH concentrations are highest over tropics, therefore, it is expected that the difference in OH from varied fields is largest over tropics. This could explain the larger posterior flux range in tropics for Inv1.

**Response: We added in the 3.1.2 (L407-415):**

**" The uncertainties in the top-down estimated regional emissions are not only due to inter-model differences of the regional OH fields but also rely on the distribution of the surface observations used in the inversions. Over the regions with large prior emissions but less constrained by observations (e.g. South America, South Asia, and China), our OH analysis leads to larger uncertainties than regions that are well constrained by observations (e.g. the North America and Canada) (Fig. S3). The results may indicate that on the regional scale, the top-down estimated CH₄ emissions and the uncertainties lead by OH are specific to the observation system retained. If more surface observations (e.g. in the southern hemisphere) or satellite columns with a more even global coverage were included in our inversions, spatial patterns of the top-down estimated CH₄ emissions and their uncertainties (as shown by Fig.3) could be different."**

**Comments:** 38-40: The sentence reads awkward. Physically, increases in OH burden cannot contribute to increases in emissions. Clarify or rephrase to avoid any confusion.

**Response: We changed the sentence to (L37-L38):**

**"From the early to the late 2000s, the optimized $CH_4$ emissions increased by $21.9\pm5.7$Tg yr$^{-1}$ (16.6-30.0Tg yr$^{-1}$), of which ~25% (on average) offsets the 0.7% (on average) increase in OH burden "**

**Comments:** 53-54: The word "additional" is confusing here.

**Response: We removed "additional"**

**Comments:** 71-72: Unclear what "catalytic chemistry" in this sentence is referred to. Also, the statement "a small perturbation of OH can result in significant change in atmospheric CH4" is inaccurate or ambiguous. The author may want to say ". . . significant change in the budget (or budget imbalance) of atmospheric CH4".

**Response: We rephrased the sentence as suggested:" A small perturbation of OH can result in significant changes in the budget of atmospheric $CH_4$ (Turner et al., 2019)."**

**Comments:** 72-75: There are other OH sources such as O3+HO2, H2O2 photolysis, and OVOCs photolysis that become important depending on the chemical environment, for example, see Lelieveld et al. (2016).

**Response: We changed in the text (L70-L75):**

**" At the global scale, tropospheric OH is mainly produced by the reaction of excited oxygen atoms (O($^1$D)) with water vapor (primary production) but also by the reaction of nitrogen oxide (NO) and ozone (O$_3$) with hydroperoxyl radicals (HO$_2$) and organic peroxy radicals (RO$_2$) (secondary production). At regional scales, photolysis of hydrogen peroxide and oxidized VOC photolysis can be important depending on the chemical environment (Lelieveld et al. 2016)."**

**And we added in the reference list:**

**Lelieveld, J., Gromov, S., Pozzer, A., and Taraborrelli, D.: Global tropospheric hydroxyl distribution, budget and reactivity, Atmospheric Chemistry and Physics, 16, 12477-12493, 10.5194/acp-16-12477-2016, 2016.**

**Comments:** 78: A direct measurement of OH is challenging but possible. But estimates of global ˘ mean from sparse direct measurements is nearly impossible because the large variation of OH as a result of its short lifetime.

**Response: We rephrased the sentence to (L77-L89) "Tropospheric OH has a very short lifetime of a few seconds (Logan et al., 1981; Lelieveld et al., 2004), hindering estimates of global OH concentrations ([OH]) through direct measurements and limiting our ability to estimate the global CH₄ sink."**

**Comments:**101-103:Optimizations of CH4 emissions together with OH concentrations have been done using 3-D model inversions (e.g., Cressot et al., 2014, Zhang et al., 2018 and Maasakkers et al., 2019), in addition to two-box model analysis. These studies all used satellite data though.

**Response: We added in the text (L105-L111):**

**"The role of OH variations on the top-down estimates of CH₄ emissions has been evaluated using two box-model inversions with surface observations (e.g. Rigby et al., 2017; Turner et al., 2017, Naus et al., 2019) and 3D models that optimize CH₄ emissions together with [OH] by assimilating surface observations (Bousquet et al., 2006) or satellite data (Cressot et al., 2014, McNorton et al., 2018; Zhang et al., 2018; Maasakkers et al., 2019). The proxy-based constraints usually optimize [OH] on a global or latitudinal scale, the impact of OH vertical and horizontal distributions being less quantified to date. Also, proxy methods do not allow to access underlying processes as direct chemistry modeling (Zhao et al., 2019). "**

**Comments**: Line 155: What temperature field do you use to compute [OH]GM-CH4 for different models? And how "troposphere" is defined in this calculation? Line 158: Is latitudinal distribution of OH also a factor (and maybe even more important factor) that results in [OH]GM-CH4 > [OH]GM-M ?

**Response: We clarified in the text (L167-L169):**

**"The tropopause height is assumed at 200hPa following Naik et al. (2013) and the 3D temperature field used to compute $[OH]_{GM-CH4}$ is from ERA Interim re-analysis meteorology data (Dee et al, 2011). "**

**As we can see in Table 1, if MOCAGE and SOCOL3 OH fields are excluded, differences between $[OH]_{GM-M}$ and $[OH]_{GM-CH4}$ are largely reduced. We clarified in the text (L175-L177):**
**" This is mainly because MOCAGE and SOCOL3 OH fields show much higher [OH] near the surface than in the upper troposphere (Zhao et al., 2019).", and we removed: " as some of the OH fields show distinct vertical distributions".**

**Comments:** Eq. 1. The (x-xb) term is repeated twice.

**Response: Thank you very much for pointing out this, we removed the (x-xb).**

**Comments:** Line 172: Since only emissions are optimized in the inversion, it's a bit misleading to say H(x) represents sensitivity to sinks.

**Response: We removed the "sinks" as suggested.**

**Comments:** Line 205: What about Cl?

**Response: We added in the text (L225-226) "The $CH_4$ sink by reaction with chlorine is not considered in our LMDz model simulations."**

**Comments:** Line 231-232: "To separate the influence of OH spatial distributions from that of global mean [OH]: : :". As commented above, it is unclear whether the OH fields vary monthly or annual mean. If the former, then in addition to influence of spatial distribution, the influence of seasonal variation is also embedded. If the latter, then the study design has a major flaw because the latitudinal distribution of OH has a pronounced seasonal cycle.

**Response: OH fields vary monthly in our inversions, the seasonal variations of OH fields can impact inversion results. Thank you for mentioning the role of the OH seasonal cycle, which is not detailed in our analysis. We clarified in the text (in section 2.2 - L186):**

**" We conduct an ensemble of variational inversions … but different prescribed monthly mean OH fields as described in Sect. 2.1."**

**In section 2.3, to emphasize the impact of OH seasonal variation, although not analyzed separately in this work. we added:**

**L254-255: "To separate the influence of OH spatial distributions (including their seasonal variations) from that of the global annual mean [OH]."**

**L257-258: "As such, Inv2 provides the uncertainty range of $CH_4$ emissions induced by OH spatial distribution in both horizontal and vertical directions as well as seasonal variations…"**

**Comments:** Line 239-240: Please denote inv3 and inv4 explicitly after 2007-2009 and 2000-2002 to

make it easier to follow.

**Response: This has been changed as suggested.**

**Comments:** Line 241-243: I don't think Inv4-Inv2 represents the impact of OH spatial distribution.

**Response: Here we mean the difference in Inv4－Inv2 estimated by different OH fields represents the uncertainties lead by the different OH spatial and seasonal distributions since they are all using OH fields scaled to the same value globally for 2000-2002.**

**We clarified in the text (L268-272):**

**" Therefore, the difference Inv3－Inv2 reveal the impact of OH on $CH_4$ emission changes between the early and late 2000s (the yellow box with solid lines of Fig. 1), Inv3－Inv4 separates the impact of OH interannual variations, and the difference Inv4－Inv2 allows assessing the uncertainties of optimized $CH_4$ emission changes due to different OH spatial and seasonal distributions (the yellow boxes with dashed lines in Fig. 1). "**

**Comments:** Line 250: Which one has the largest trend, which may be more relevant in this setting?

**We added in the text (L279):**

**"…shows the largest year-to-year OH variations and a positive trend of 0.35% $yr^{-1}$ …"**

**Comments:** Line 269-273: This sentence does not flow smoothly within the context (results from Inv1). Remove it or move it somewhere else.

**Response: We removed this sentence as suggested**

**Comments:** Line 278: Not clear to me how this helps decreasing discrepancies with bottom-up estimates? Fig. 2 does not show the discrepancies are reduced to me. Please clarify. Also please provide the values (and ranges) of bottom-up estimates in the text for a clear comparison.

**Response: We removed "help decreasing discrepancies with bottom-up estimations", and we added the number in the text (L316-L320):**

**"The minimum-maximum range of the $CH_4$ emissions estimated by the 10 OH fields is almost similar to the range estimated by previous bottom-up studies (542-852Tg $yr^{-1}$ given by Kirschke et al., 2013 and 583-861Tg $yr^{-1}$ given by Saunois et al, 2016) from GCP syntheses and much larger than that reported by an ensemble of top-down studies for 2000-2009 in Kirschke et al. (2013) (526-**

569Tg yr$^{-1}$), Saunois et al. (2016) (535-566Tg yr$^{-1}$) or the recent Saunois et al. (2019) (522-559 Tg yr$^{-1}$). (Table 2 and Fig. 2). "

**Comments:** Line 284-290: Is it possible that the difference is due to the fact that [OH]GM-CH4 is used here instead of [OH]GM-M (which I assume was used in these studies)? It does not convince me that the difference is due to the inter-hemispheric transport and stratospheric loss in 3-D model vs. 2-box model. Choices of hemispheric mean reaction rate of OH+CH4 can also introduce biases in 2-box model.

**Response: For the two-box model inversion, the [OH] GM-CH4 is the same as [OH] GM-M since the air mass and temperature are homogeneously distributed over space. For 3D model inversion, the optimized CH$_4$ emissions do not show a linear relationship with [OH]GM-M. One can see that the [OH]GM-M of CMAM OH field (11.3×10$^5$ molec cm$^{-3}$) is a bit lower than that EMAC-L90MA (11.5 ×10$^5$ molec cm$^{-3}$ ) and CESM1-WACCM (11.4×10$^5$ molec cm$^{-3}$), but the top-down estimated CH4 emissions using CMAM OH field (599Tg yr$^{-1}$) is higher than that estimated using CESM1-WACCM (578Tg yr$^{-1}$) and EMAC-L90MA (589Tg yr$^{-1}$).**

**For the explanation of the difference between two-box model and 3-D model inversions, we agree that the choice of hemispheric mean rate is a more important factor. We added in the text (L342-L345):**

**"This difference probably results from the different hemispheric mean reaction rates of OH+CH$_4$ applied in box models, but could also be due to different treatments of inter-hemispheric transport and stratospheric CH$_4$ loss in global 3D transport models compared to simplified box-models (Naus et al., 2019)."**

**Comments:** Line 316: Does the seasonality of OH fields also play a role here?

**Response: Yes, the seasonality can also contribute to the differences in Inv2. As we cannot separate the contribution from seasonal variations and spatial distribution, we emphasized this in Section 2.3 (L245-L255):**

**"To separate the influence of OH spatial and seasonal distributions from that of the global mean [OH]."**

**Comments:** Line 338: Please explicitly state which uncertainty sources Saunois et al. (2016) considered. The comparison may be misleading otherwise.

**Response: We clarify in the text (L387-L391):**

**"The uncertainties in global OH burden and distributions lead to larger uncertainty (maximum－minimum) in top-down estimated $CH_4$ emissions over the tropics (>20% of multi-inversion mean) and smaller uncertainty over the northern mid-latitude regions (14%) compare with that lead by transport model errors and different observations given by Saunois et al. (2016) (13% over tropics and 20% over northern mid-latitude regions) (Table 3)."**

**Comments:** Line 343-345: Likely because these regions have high prior emissions, but are not well constrained by surface measurements. So, it should be stated that these regional features are not intrinsic of the atmosphere, but specific to the observing system of interest.

**Response: As already mentioned in the first comments, we added in the text (L407-415):**

**"The uncertainties in the top-down estimated regional emissions are not only due to inter-model differences of the regional OH fields but also rely on the distribution of the surface observations used in the inversions. Over the regions with large prior emissions but less constrained by observations (e.g. South America, South Asia, and China), our OH analysis leads to larger uncertainties than regions that are well constrained by observations (e.g. the North America and Canada) (Fig. S3). The results may indicate that on the regional scale, the top-down estimated $CH_4$ emissions and the uncertainties lead by OH are specific to the observation system retained. If more surface observations (e.g. in the southern hemisphere) or satellite columns with a more even global coverage were included in our inversions, spatial patterns of the top-down estimated $CH_4$ emissions and their uncertainties (as shown by Fig.3) could be different."**

**Comments:** Line 355: what is the "total differences"? How large are they?

**Response: We clarified in the text (L427-L429):**

**"… account for 50% of the differences due to both OH burden and spatial distributions… "**

**Comments:** Line 364-367: I don't understand the logic here. I think it is probably related to OH concentration being much higher in tropics than extra-tropics.

**Response: When scaling all OH fields to the same total global loss, the inter-model difference of OH is reduced by 33% over northern mid and high latitudes and uncertainties in top-down estimated CH$_4$ emissions are reduced by only 22%. Over northern tropical regions, the inter-model difference in OH is reduced by 67% but the uncertainties in CH$_4$ emissions are reduced by 93%, as we show in the text. The explanations here, we think, are similar to the comments for Line 343-345, which related that OH over tropical regions is more sensitive to global OH burdens as less constrained by local/direct observations. We clarified this point in the text (L434-L440):**

**"Over tropical regions, CH$_4$ emissions are less constrained (with few to none observation sites near source regions) than in the northern extra-tropics, where several monitoring sites located at or near the regions with high CH$_4$ emission rates and high OH uncertainties (e.g. North America, Europe, and downwind of East Asia). Thus, CH$_4$ emissions over the tropical regions mainly contribute to match the global total CH$_4$ sinks (instead of the sinks over the tropical regions only) estimated by inversion systems. When all OH fields are scaled to the same CH$_4$ losses (Inv2), differences of emissions over the tropical regions are therefore largely reduced. "**

**Comments:** Line 379: The range of global total CH4 emissions by Inv2 (551+-2Tg a-1) should be reported and discussed in 3.3.1, in comparison with Inv1.

**Response: We added in section 3.1.1 (L347-L350):**

**" With the OH fields scaled to the same [OH]$_{GM\text{-}CH4}$ (11.1$\times$10$^5$molec cm$^{-3}$ ), the Inv2 simulations (assuming a global total OH burden well constrained) estimated global CH$_4$ emissions of 551$\pm$2Tg yr$^{-1}$ (Table 3), as expected by the scaling. Differences in OH spatial distributions only lead to negligible uncertainty in global total CH$_4$ emissions estimated by top-down inversions."**

**Comments:** Line 418: Be clearer what "global scale increase" in this sentence is referred to. It is ambiguous in the current form.

**Response: We add in the text (L491):" the increase in global mean [OH]"**

**Comments:** Line 485-489: The assessment of the uncertainty due to OH fields relative to other uncertainty sources are too qualitative throughout the manuscript. More insight can be gained by quantitatively comparing to uncertainty estimates in literature such as Saunois et al.

**Response: We added in section 3.2.2(L555-L566):**

**" We now compare the uncertainty of top-down estimated CH$_4$ emission changes from the early to the late 2000s due to different OH spatial-temporal variations with that ensemble of top-down studies given by Saunois et al. (2017). For the sectoral emissions, the emission changes from agriculture and waste and from wetland show the largest uncertainties (more than 50% of multi-inversions mean, Inv3－Inv2 in Table 6) induced by OH spatial-temporal variations, comparable to that given by Saunois et al. (2017). On the contrary, the uncertainty of fossil fuel emission changes (24% of multi-inversions mean) is much smaller than that given by Saunois et al. (2017). For regional CH$_4$ emission changes, the uncertainty induced by OH spatial-temporal variations is usually larger than the multi-inversion mean emission changes (except South Asia) and similar to that given by Saunois et al. (2017). The large differences existing in different top-down estimated regional and sectoral emission changes are mainly attributed to model transport errors in Saunois et al. (2017). Here, our results show that uncertainties due to OH spatio-temporal variations can lead to similar biases in top-down estimated CH$_4$ emission changes."**

**Comments:** Table 2 Quite confusing. Why global and hemispheric emissions are only shown for Inv1, but the inter-hemispheric differences are shown for both Inv1 and Inv2? Also, unit should be denoted in the caption.

**Response: We added the global and hemispheric CH$_4$ emissions estimated by Inv2 to Table2 as suggested. And we included the unit.**

**Table 2.** The global total, hemispheric CH$_4$ emissions, and inter-hemispheric difference of CH$_4$ emissions calculated by Inv1 and Inv2 during the early 2000s (2000/07/01-2002/06/01) in Tg yr$^{-1}$.

| Unit: Tg yr$^{-1}$ | Inv1 original OH | | | | Inv2 scaled OH | | | |
|---|---|---|---|---|---|---|---|---|
| | Global | 0-90 °N | 90 °S-0 | N-S$_{Inv1}$ | Global | 0-90 °N | 90 °S-0 | N-S$_{Inv2}$ |
| **Prior** | 522 | 384 | 138 | 246 | 522 | 384 | 138 | 246 |
| **TransCom** | 530 | 368 | 162 | 206 | 549 | 377 | 172 | 205 |
| **INCA NMHC-AER-S** | 518 | 380 | 138 | 242 | 553 | 399 | 154 | 245 |
| **INCA NMHC** | 552 | 392 | 160 | 232 | 552 | 392 | 160 | 232 |
| **CESM1-WACCM** | 587 | 420 | 166 | 254 | 551 | 400 | 151 | 249 |
| **CMAM** | 599 | 419 | 180 | 239 | 553 | 399 | 154 | 245 |
| **EMAC-L90MA** | 589 | 414 | 175 | 239 | 555 | 396 | 159 | 237 |
| **GEOSCCM** | 611 | 424 | 187 | 237 | 550 | 392 | 159 | 233 |
| **MOCAGE** | 716 | /[a] | / | / | / | / | / | / |
| **MRI-ESM1r1** | 553 | 396 | 156 | 240 | 548 | 396 | 152 | 244 |
| **SOCOL3** | 757 | / | / | / | / | / | / | / |

| Mean±SD | 601±78 | 401±21 | 166±15 | 236±14 | 551±2 | 393±7 | 158±7 | 236±14 |

[a] We do not analyze the hemispheric $CH_4$ emission estimated with MOCAGE and SOCOL3 OH field since inversions using the two OH fields calculate much higher $CH_4$ emissions than using other OH fields.

**Comments:** Table 5 With fixed OH field, you still expect an increasing OH sink (and therefore increasing emissions) because of increasing CH4 concentration and temperature. This should be clarified somewhere in the text.

**Response: We clarified in the 3.2.1 (L522-L525):**

**"Keeping OH fields from 2000-2002, top-down estimated $CH_4$ emissions increase by $16.9\pm1.9$Tg yr$^{-1}$ (14.3-19.3Tg yr$^{-1}$, Table 5) between the early 2000s (Inv2) to the late 2000s (Inv4) in response to increasing atmospheric $CH_4$ mixing ratios and temperature. This represents 75% of total optimized emission changes (Inv3－inv2) between the early and late 2000s ($21.9\pm5.7$Tg yr$^{-1}$, Table 5)."**

**Comments:** Fig. 2 The R2=0.99 line in the right panel: it should be acknowledged that other sinks of methane (such as soil absorption, Cl, and stratospheric loss) are not optimized and are specified with the same field in these inversions. Uncertainty in these sinks, if considered, will certainly create some spread in the data.

**Response: We added in the Section 3.1.1 (L334-338): " Where a $1\times10^5$ molec cm$^{-3}$ (1%) increase in $[OH]_{GM\text{-}CH4}$ will increase the top-down estimated $CH_4$ emissions ($EMIS_{CH4}$) by 40.4 Tg yr$^{-1}$, consistent with that given by He et al. (2020) using full-chemistry modeling and a mass balance approach. Other $CH_4$ sinks including soil uptake and oxidation by $O^1(D)$, which are prescribed in this study, remove 66.7Tg yr$^{-1}$ $CH_4$. If uncertainties in all the $CH_4$ sinks were also considered, the correlation between optimized $CH_4$ emissions and the $[OH]_{GM\text{-}CH4}$ would be reduced. "**

**Comments:** Fig. 3 To interpret this figure, the author should consider the uneven sampling of the surface network. The ranges of inferred regional emissions are large where observations are sparse, because it "costs" the least for the inversion to adjust in these regions. The inference for regional emissions is specific to the particular observing system. Having more surface stations in the southern hemisphere, or including satellite observations, would change the spatial pattern shown in this figure.

**Response: As stated in previous comments, we added in section 3.1.2 (L407-L415):**

**"The uncertainties in the top-down estimated regional emissions are not only due to inter－model differences of the regional OH fields but also rely on the distribution of the surface observations**

used in the inversions. Over the regions with large prior emissions but less constrained by observations (e.g. South America, South Asia, and China), our OH analysis leads to larger uncertainties than regions that are well constrained by observations (e.g. the North America and Canada) (Fig. S3). The results may indicate that on the regional scale, the top-down estimated $CH_4$ emissions and the uncertainties lead by OH are specific to the observation system retained. If more surface observations (e.g. in the southern hemisphere) or satellite columns with a more even global coverage were included in our inversions, spatial patterns of the top-down estimated $CH_4$ emissions and their uncertainties (as shown by Fig.3) could be different. ”

---

## Author Comment (AC3) · 12 May 2020

*Reply to RC2: ' Review'*

1 Overview:

Review of "Influences of hydroxyl radicals (OH) on top-down estimates of the global and regional methane budgets" by Zhao et al. This review slipped through the cracks as the COVID-19 situation evolved here. My sincere apologies for any hold ups. Zhao et al. present an analysis of a set of methane inversions using a set of 10 different OH fields from the CCMI experiment. They find the magnitude of methane emissions differs by roughly 30% (518-757 Tg/yr) depending on what OH fields they use. Over all, the study is useful in quantifying some of the uncertainties in methane emission estimates due to uncertain OH concentrations. The main shortcoming is the lack of discussion of what actually causes some of the differences (or really any discussion of OH). The figures are high quality but the text is very hard to follow because it's filled with many acronyms and parenthetical expressions. I would recommend major revisions.

**Response: We thank the reviewer for his/her helpful comments. This manuscript is the second step of our previous study, in which we have made a detailed description of the CCMI, INCA and TransCom OH fields and where we analyzed the factors controlling the inter-model differences in OH burden and spatial distribution and the increasing trend of OH simulated by CCMI models (Zhao et al., 2019). This manuscript mainly focuses on the impact of the inter-model differences of different OH fields on the top-down estimates of $CH_4$ emissions. Thus, regarding discussions on OH, we directly refer to the conclusion from Zhao et al. (2019) (see the response for the comments 2.1) without re-detailing all results. However, we now better explain the link with our first paper in the revised version and include summaries of results from Zhao et al. (2019) in this second paper.**

**Also, we have rephrased large parts of the original manuscript in this revised version, and especially**

**Section 3.2, which includes most of the acronyms and parenthetical expressions. All of the other comments have been addressed in the revised manuscript. Please see out itemized responses below.**

2 Comments:

2.1 What processes actually drive these differences?

The main issue I feel is totally missing from the manuscript is any discussion of what processes are actually driving some of these differences. I believe there was only a single paragraph (Lines 70-79) even mentioning anything about what might affect OH. For example, do some of the CCMI models or inversions show consistent patterns with known climate oscillations? This was surprising given that this is a paper focused on how OH impacts methane. The obvious question is what leads to these differences/similarities in OH. The authors seem to treat the CCMI models as a black box which makes it hard to gain any understanding of what's happening.

**Response:**

**As mentioned in the overview, this paper follows Zhao et al. (2019) where we analyzed in detail OH fields from CCMI models. Including the main conclusions of this first paper in the updated section 2.1 of this paper provides the required elements on what causes OH differences in CCMI models.**

**In the introduction, to clarify the link with our previous paper, we added (L112-121):**

  **"This paper follows the work of Zhao et al. (2019), where we analyzed in details 10 OH fields derived from atmospheric chemistry models considering different chemistry, emissions, and dynamics (Patra et al., 2011; Szopa et al., 2013; Hegglin and Lamarque, 2015; Morgenstern et al., 2017; Zhao et al., 2019; Terrenoire et al., 2019). We now aim to build on this previous paper to estimate the impact of these OH fields on methane emissions as inferred by an atmospheric 4D variational inversion system. To do so, we use each of the OH fields in the 4D variational inversion**

system PYVAR-LMDz based on LMDZ-SACS (Laboratoire de Météorologie Dynamique model with Zooming capability-Simplified Atmospheric Chemistry System) 3D chemical transport model to evaluate the influence of OH distributions and variations on the top-down estimated global and regional CH4 budget. Section 2 briefly describes the OH fields and their characteristics and underlying processes (see also Zhao et al., 2019 for more details)"

In section 2.1, we added (L146-L154):

"The inter-model differences of OH burden and vertical distributions are mainly attributed to differences in chemical mechanisms related to NO production and loss. The differences in [OH] spatial distributions are due to applying different natural emissions: for example, primary biogenic VOC emissions and NO emissions from soil and lightning (Zhao et al., 2019). As a result, the regions dominated by natural emissions (e.g. South America, central Africa) show the largest inter-model differences in [OH] (Fig.S1). The CCMI models consistently simulated positive OH trend during 2000-2010, mainly due to more OH production by NO than loss by CO over the East and Southeast Asia and positive trend of water vapor over the tropical regions (Zhao et al., 2019; Nicely et al., 2020). More details can be found in Zhao et al. (2019) and the herein cited literature."

In section 3.1.1 (L310-L314), we added:

"The high [OH]$_{GM-CH_4}$ simulated by SOCOL3 and MOCAGE are mainly due to high surface and mid-tropospheric NO mixing ratio simulated by these two models (Zhao et al., 2019). As analyzed in Zhao et al. (2019), the lack of $N_2O_5$ heterogeneous hydrolysis (by both SOCOL3 and MOCAGE) and the overestimation of tropospheric NO production by $NO_2$ photolysis (by SOCOL3) are the major factors behind the overestimation of NO and OH."

Given this, the only major take-away I had from the paper is that "OH can lead to big differences in

methane estimates", but this was already demonstrated by the box modeling papers (and others) that Zhao et al. are highly critical of. For example, the Rigby paper had error bars on their OH fields that bounded zero and the Turner paper had a case where OH didn't change. Both of these led to radically different methane emissions.

**Response: Two-box models are an effective tool to assess changes in global CH$_4$ budget (Rigby et al., 2017, Turner et al., 2017), but we think that 3D analysis is still needed : (i) to check if box-model results are not biased by the oversimplification of atmospheric transport, (ii) to infer the regional CH$_4$ budgets, and (iii) to estimate methane decadal budgets as box-models are less relevant for estimating budgets than budget changes (Saunois et al., 2019). The study aims to quantify the uncertainties lead by using the prescribed OH fields in 3D model inversions.**

**Also, we have been more precise in the abstract and conclusions for the reader to get a more complete take away of our work (regions that are sensitive/important, quantitative estimates, comparison with other causes of uncertainties).**

**We added in the abstract (L25-L27):**
**"Current top-down estimates of the global and regional CH$_4$ budget using 3D models usually apply prescribed OH fields and attribute model-observation mismatches almost exclusively to CH$_4$ emissions, leaving the uncertainties due to prescribed OH field less quantified. "**
**And we clarified in the text that we conducted inversions using an ensemble of OH fields not only to show that OH can lead to differences in CH$_4$ emissions but also to quantify the influences at global and regional scales.**

**In "conclusions and discussion", we compared the uncertainties lead by OH with other causes of uncertainty.**

**For global total CH$_4$ emissions (L640-L644):**

" Based on the ensemble of 10 original OH fields ([OH]$_{GM-CH4}$:10.3-16.3$\times10^5$ molec cm$^{-3}$), the global total CH$_4$ emissions inverted by our system vary from 518 to 757Tg yr$^{-1}$ during the early 2000s, similar to the CH$_4$ emission range estimated by previous bottom-up syntheses and larger than the range reported by the top-down studies (Kirschke et al., 2013; Saunois et al, 2016). "

**For the regional emissions (L647-L654):**

" At regional scale (excluding the two highest OH fields), CH$_4$ emission uncertainties due to different OH global burdens and distributions are largest over South America (37% of multi-inversion mean), South Asia (24%), and China (39%), resulting in significant uncertainties in optimized emissions from the wetland and agriculture and waste sectors. These uncertainties are comparable in these regions with those due to model transport errors and inversion system set-up (Locatelli et al., 2013; Saunois et al., 2016). For these regions, the uncertainty due to OH is critical for understanding their methane budget. In other regions, OH leads to smaller uncertainties compared to that given by Locatelli et al. (2013) and Saunois et al. (2016)."

**And we emphasized the importance of the OH spatial distributions on the top-down estimation of regional CH4 budget (L705-713):**

" Our results indicate that OH spatial distributions, which are difficult to obtain from proxy observations (e.g. MCF), are equally important as the global OH burden for constraining CH$_4$ emissions over mid- and high-latitude regions. Constraining global annual mean OH based on proxy observations (e.g. Zhang et al., 2018; Maasakkers et al., 2019) provides a constraint on global total methane emissions, through the necessity of balancing the global budget (sum of source – sum of sinks = atmospheric growth rate). It also largely reduces uncertainties in optimized CH$_4$ emissions due to OH over most of the tropical regions but not over South America and overall mid-

**high latitude regions. Also, the spatial and seasonal distributions of OH is found critical to properly infer temporal changes of regional and sectoral CH₄ emissions."**

Back to my point, I would find this manuscript much more useful and compelling if the authors actually highlighted processes and phenomena that lead to similar methane inversion responses. From Holmes et al., ACP (2013; https://doi.org/10.5194/acp-13- 285-2013) we know some of the major processes that influence OH and Turner et al., PNAS (2018; https://doi.org/10.1073/pnas.1807532115) showed how this can with things like ENSO, do the authors see ENSO signals in the methane inversions? A recent paper from Nguyen et al., GRL (2020) tried to look at these feedbacks in a simple model. The authors should at least touch on the processes that influence OH, particularly those that could also influence methane.

**Response: We acknowledge that it is important to analyze processes and phenomena that lead to similar methane inversion responses. During the time period of this manuscript (the 2000s), the CCMI model simulated OH show a consistent positive trend (Zhao et al., 2019; Nicely 2020). As stated in our response to comments 2.1, the positive OH trend is mainly due to more OH production by NO than loss by CO over the East and Southeast Asia and positive trend of water vapor over the tropical regions (Zhao et al., 2019; Nicely et al., 2020). We have analyzed the impact of positive [OH] trend during the 2000s on top-down estimates of CH₄ emissions in section 3.2 and section 3.3. The ENSO signal during the early 2000s is very weak (with small year-to-year variations of [OH]) and the time period of this paper very short. Therefore, analyzing the impact of ENSO seems beyond the scope of this paper. However, please note that we have another manuscript submitted to ACP using CCMI models that analyzes the impact of trend and interannual variability of OH on the CH₄ budget on the decadal time scale (1980-2010) with a focus on the ENSO (https://doi.org/10.5194/acp-2020-308).**

**Yet, we discussed shortly the impact of ENSO in Section 4 (L692-L701):**

**"The trend and interannual variations of tropospheric OH burden are determined by both precursor emissions from anthropogenic and natural sources and climate change (Holmes et al., 2013; Murray et al., 2014). Based on satellite observations, Gauber et al. (2017) estimated that ~20% decrease in atmospheric CO concentrations during 2002-2013 led to an ~8% increase in atmospheric [OH]. The El Niño-Southern Oscillation (ENSO) has been proven to impact the tropospheric OH burden and CH4 lifetime mainly through changes in biomass burning from CO (Nicely et al., 2020; Nguyen et al, 2020) and in NO emission from lightning (Murrary et al., 2013; Turner et al., 2018). The ENSO signal is weak during the early 2000s, resulting in small interannual variations of tropospheric OH burden (Zhao et al., 2019). The mechanisms of OH variations related to ENSO and their impacts on the CH4 budget need to be explored by inversions, but over a longer time period than this study (e.g. 1980-2010, Zhao et al., 2020)."**

2.2 Oversight of previous work and faith in the CCMI models

The authors seem to have quite a bit of faith in the CCMI models, more than this reviewer finds to be justified. There are quite a few known shortcomings of the models. For example, the models don't even get the ratio of the N/S gradient in OH correct. Yet the authors are quick to criticize MCF-constrained [OH] fields with seemingly no validation of their own OH fields (e.g., Lines 595 600). Is their analysis consistent with MCF? The authors seem to be arguing that these model derived forward simulations of OH are more reliable than reconstructions. The strongest claims made in this paper seem to be those that are critical of previous work estimating OH (e.g., Rigby and Turner). For example, Lines 595-600, the abstract is dismissive of box modeling results: 'previous research mostly relied on box modeling inversions with a very simplified atmospheric transport". The latter line in the abstract isn't even correct as there has been quite a bit of non-box model work the authors seem to discount or miss: McNorton et al., ACP (2016; https://doi.org/10.5194/acp-16- 7943-2016), Gaubert et al., GRL (2017;

https://doi.org/10.1002/2017GL074987), Rigby et al., PNAS (2017; https://doi.org/10.1073/pnas.1616426114), Turner et al., PNAS (2017; https://doi.org/10.1073/pnas.1616020114), McNorton et al., ACP (2018; https://doi.org/10.5194/acp-18-18149-2018), Maasakkers et al., ACP (2019; https://doi.org/10.5194/acp-19-7859-2019), Naus et al., ACP (2018; https://doi.org/10.5194/acp 19-407-2019), Nguyen et al., (2020; https://doi.org/10.1029/2019GL085706), and He et al., ACP (2020; https://doi.org/10.5194/acp-20-805-2020). About half of these papers use 3-D atmospheric transport models and some even include fully-coupled chemistry (e.g., He et al., 2020), which is more comprehensive than the models used by the authors. The authors should do a more complete reading of the literature as they don't cite Holmes et al., ACP (2013), Murray et al., ACP (2014), or any of Michael Prather's papers.

**Response:**

**About CCMI models and proxy methods.**

**To date, we have mostly two approaches to estimate regional and global [OH], one using direct atmospheric chemistry modeling and one using proxy tracers, the main one being MCF. We do not have a specific faith in CCMI models (neither we deny the interest of MCF) but here, we choose to investigate the first approach using chemistry models, taking benefit of an important collaborative effort of this scientific community (CCMI experiments) to compare and, in fine, improve their models. Each method has its caveats and, following the comment of the reviewer, we try to balance more things between the two approaches in the updated version of the manuscript.**

**We added in the introduction(L104-106): "However, the OH fields simulated by atmospheric chemistry models show some uncertainties in both global burden and spatial-temporal variations (Naik et al., 2013; Zhao et al., 2019)"**

For the N/S ratio>1 simulated by CCMI models, we reported their ranges, with a clarification in the text (L176-179):

" The inter-hemispheric OH ratios range from 1.0 to 1.5, larger than ones derived from MCF inversions (e.g. Bousquet et al., 2005; Patra et al., 2014), partly explained by the underestimation of CO in the northern hemisphere by atmospheric chemistry models (Naik et al., 2013)."

For lines 595-600, we just wanted to mention that the OH trend simulated by CCMI models (positive) is different than that from MCF inversions, which mainly show a decrease of [OH] after the early 2000s. It is hard to say which one is correct since both of the methods have their caveats regarding trends. For the increasing trend simulated by CCMI models, we have discussed the impact on $CH_4$ budget in the manuscript by writing with caution:" if the CCMI models represent the OH trend properly, a higher increasing trend of $CH_4$ emissions is needed to match the CH4 observations (compared to the $CH_4$ emission trend derived using constant OH).". We do not argue that CCMI models simulate a more realistic OH trend than two-box model inversions and/or proxy-based methods.

For the abstract, we removed "previous research mostly relied on box modeling inversions with a very simplified atmospheric transport".

About missing literature

For previous studies that quantify the impact of OH variations on the top-down estimate of CH4 by 3D models, we acknowledge the missing references and thank the reviewer to have provided them. We added in the introduction (L105-L112):

"The role of OH variations on the top-down estimates of $CH_4$ emissions has been evaluated using

two box-model inversions with surface observations (e.g. Rigby et al., 2017; Turner et al., 2017, Naus et al., 2019) and 3D models that optimize $CH_4$ emissions together with [OH] by assimilating surface observations (Bousquet et al., 2006) or satellite data (Cressot et al., 2014, McNorton et al., 2018; Zhang et al., 2018; Maasakkers et al., 2019). The proxy-based constraints usually optimize [OH] on a global or latitudinal scale, the impact of OH vertical and horizontal distributions being less quantified to date. Also, proxy methods do not allow to access underlying processes as direct chemistry modeling (Zhao et al., 2019). "

He et al. (2020) estimated the global $CH_4$ budget by forward-model simulations and mass balance method and estimated that a 1 % change in OH levels could lead to an annual mean difference of $\sim 4$ Tg $yr^{-1}$ in the optimized emissions, consistent with our top-down estimates. We cited in Section 3.1.1 (L334-336):
"Where a $1 \times 10^5$ molec $cm^{-3}$ (1%) increase in [OH]$_{GM-CH4}$ will increase the top-down estimated $CH_4$ emissions (EMIS$_{CH4}$) by 40.4 Tg $yr^{-1}$, consistent with that given by He et al. (2020) using full-chemistry modeling and a mass balance approach."

Holmes et al. ACP (2013), Murray et al. ACP (2014), Gaubert et al. (2017), Nguyen et al. (2020) analyzed the factors controlling OH variability, we cited them in the conclusions and discussion (L692-L698):
"The trend and interannual variations of tropospheric OH burden are determined by both precursor emissions from anthropogenic and natural sources and climate change (Murray et al., 2014; Holmes et al., 2013). Based on satellite observations, Gauber et al. (2017) estimated that ~20% decrease in atmospheric CO concentrations during 2002-2013 led to an ~8% increase in atmospheric [OH]. The El Niño-Southern Oscillation (ENSO) has been proven to impact the tropospheric OH burden and $CH_4$ lifetime mainly through changes in biomass burning from CO

(Nicely et al., 2020; Nguyen et al, 2020) and in NO emission from lightning (Murrary et al., 2013; Turner et al., 2018)."

We also cited Prather et al. (2012) and Prather et al. (2017):

L85: "resulting in a chemical lifetime of ~9 years for tropospheric $CH_4$ (Prather et al., 2012; Naik et al., 2013)."

L86-L88: "However, accurate estimation of [OH] magnitude, distributions, and year-to-year variations needed for $CH_4$ emission optimizations are still pending (Prather et al., 2017; Turner et al., 2019)."

And in the conclusions and discussion (L729-L731):

"In addition, as suggested by Prather et al. (2017), the OH inversion would benefit from including in their prior data the responses of [OH] to variations of the precursor emissions (e.g. biomass burning and lighting) using the uncertainties estimated by 3D models."

We added the following references:

McNorton, J., Wilson, C., Gloor, M., Parker, R. J., Boesch, H., Feng, W., Hossaini, R., and Chipperfield, M. P.: Attribution of recent increases in atmospheric methane through 3-D inverse modelling, Atmos. Chem. Phys., 18, 18149-18168, 10.5194/acp-18-18149-2018, 2018.

Holmes, C. D., Prather, M. J., Søvde, O. A., and Myhre, G.: Future methane, hydroxyl, and their uncertainties: key climate and emission parameters for future predictions, Atmospheric Chemistry and Physics, 13, 285-302, 10.5194/acp-13-285-2013, 2013.

Gaubert, B., Worden, H. M., Arellano, A. F. J., Emmons, L. K., Tilmes, S., Barré, J., Martinez Alonso, S., Vitt, F., Anderson, J. L., Alkemade, F., Houweling, S., and Edwards, D. P.: Chemical

Feedback From Decreasing Carbon Monoxide Emissions, Geophysical Research Letters, 44, 9985-9995, 10.1002/2017gl074987, 2017.

Nguyen, N. H., Turner, A. J., Yin, Y., Prather, M. J., and Frankenberg, C.: Effects of Chemical Feedbacks on Decadal Methane Emissions Estimates, Geophysical Research Letters, 47, e2019GL085706, 10.1029/2019gl085706, 2020.

He, J., Naik, V., Horowitz, L. W., Dlugokencky, E., and Thoning, K.: Investigation of the global methane budget over 1980–2017 using GFDL-AM4.1, Atmos. Chem. Phys., 20, 805-827, 10.5194/acp-20-805-2020, 2020.

Murray, L. T., Logan, J. A., and Jacob, D. J.: Interannual variability in tropical tropospheric ozone and OH: The role of lightning, Journal of Geophysical Research: Atmospheres, 118, 11,468-411,480, 10.1002/jgrd.50857, 2013.

Murray, L. T., Mickley, L. J., Kaplan, J. O., Sofen, E. D., Pfeiffer, M., and Alexander, B.: Factors controlling variability in the oxidative capacity of the troposphere since the Last Glacial Maximum, Atmospheric Chemistry and Physics, 14, 3589-3622, 10.5194/acp-14-3589-2014, 2014.

Prather, M. J., Holmes, C. D., and Hsu, J.: Reactive greenhouse gas scenarios: Systematic exploration of uncertainties and the role of atmospheric chemistry, Geophysical Research Letters, 39, L09803, doi:10.1029/2012GL051440, 2012.

Prather, M. J., and Holmes, C. D.: Overexplaining or underexplaining methane's role in climate change, Proceedings of the National Academy of Sciences, 114, 5324-5326, 10.1073/pnas.1704884114, 2017.

Nicely, J. M., Duncan, B. N., Hanisco, T. F., Wolfe, G. M., Salawitch, R. J., Deushi, M., Haslerud, A. S., Jöckel, P., Josse, B., Kinnison, D. E., Klekociuk, A., Manyin, M. E., Maréçal, V., Morgenstern, O., Murray, L. T., Myhre, G., Oman, L. D., Pitari, G., Pozzer, A., Quaglia, I., Revell, L. E., Rozanov, E., Stenke, A., Stone, K., Strahan, S., Tilmes, S., Tost, H., Westervelt, D. M., and Zeng, G.: A machine learning examination of hydroxyl radical differences among model simulations for CCMI-

**1, Atmos. Chem. Phys., 20, 1341-1361, 10.5194/acp-20-1341-2020, 2020.**

**Zhao, Y., Saunois, M., Bousquet, P., Lin, X., Berchet, A., Hegglin, M. I., Canadell, J. G., Jackson, R. B., Deushi, M., Jöckel, P., Kinnison, D., Kirner, O., Strode, S., Tilmes, S., Dlugokencky, E. J., and Zheng, B.: On the role of trend and variability of hydroxyl radical (OH) in the global methane budget, Atmos. Chem. Phys. Discuss., 2020, 1-28, 10.5194/acp-2020-308, 2020.**

**Other references were already cited in the previous version of the paper.**

2.3 Very difficult to follow The paper is filled with jargon and abbreviations. For example, nearly half of the text in Lines 440-452 are acronyms or parenthetical expressions interjecting things. This was very hard to follow as a reader.

**Response: the jargon and abbreviations are mainly used in the 3.2, we reduced the jargon and abbreviations used in this section and organize sentences in the text.**

---

## Author Response (AR2)

Dear Editor:

Thank you very much for the helpful corrections.

Now I round to the nearest integer for the CH$_4$ emissions and emission changes(mainly the numbers in Sect.3.2). For the emission trends(Sect.3.3), I still keep one decimal place or one significant digit (for the values<0.1) since the numbers are usually small. The modified data are highlighted in the manuscript.

Sincerely, Yuanhong Zhao et al.

[revised manuscript text omitted]